# *ImmunoCluster* provides a computational framework for the nonspecialist to profile high-dimensional cytometry data

James W Opzoomer[1†], Jessica A Timms[1†], Kevin Blighe[1†], Thanos P Mourikis[1], Nicolas Chapuis[2], Richard Bekoe[3], Sedigeh Kareemaghay[4], Paola Nocerino[1], Benedetta Apollonio[1], Alan G Ramsay[1], Mahvash Tavassoli[4], Claire Harrison[1,5], Francesca Ciccarelli[1,6], Peter Parker[1,7], Michaela Fontenay[2], Paul R Barber[1,3], James N Arnold[1‡*], Shahram Kordasti[1,5‡*]

[1]School of Cancer and Pharmaceutical Sciences, King's College London, Faculty of Life Sciences and Medicine, Guy's Hospital, London, United Kingdom; [2]Institut Cochin, Institut National de la Santé et de la Recherche Médicale U1016, Centre National de la Recherche Scientifique, Unité Mixte de Recherche 8104, Université Paris Descartes, Paris, France; [3]UCL Cancer Institute, Paul O'Gorman Building, University College London, London, United Kingdom; [4]Centre for Host Microbiome Interaction, FoDOCS, King's College, Guy's Hospital, London, United Kingdom; [5]Haematology Department, Guy's Hospital, London, United Kingdom; [6]Cancer Systems Biology Laboratory, The Francis Crick Institute, London, United Kingdom; [7]Francis Crick Institute, London, United Kingdom

**\*For correspondence:**
james.n.arnold@kcl.ac.uk (JNA);
shahram.kordasti@kcl.ac.uk (SK)

[†]These authors contributed equally to this work
[‡]These authors also contributed equally to this work

**Abstract** High-dimensional cytometry is an innovative tool for immune monitoring in health and disease, and it has provided novel insight into the underlying biology as well as biomarkers for a variety of diseases. However, the analysis of large multiparametric datasets usually requires specialist computational knowledge. Here, we describe *ImmunoCluster* (https://github.com/kordastilab/ImmunoCluster), an R package for immune profiling cellular heterogeneity in high-dimensional liquid and imaging mass cytometry, and flow cytometry data, designed to facilitate computational analysis by a nonspecialist. The analysis framework implemented within *ImmunoCluster* is readily scalable to millions of cells and provides a variety of visualization and analytical approaches, as well as a rich array of plotting tools that can be tailored to users' needs. The protocol consists of three core computational stages: (1) data import and quality control; (2) dimensionality reduction and unsupervised clustering; and (3) annotation and differential testing, all contained within an R-based open-source framework.

## Introduction

Systems immunology approaches aim to explore and understand the complexity of the immune system. However, with 350 cluster of differentiation (CD) antigens, over 100 cytokines and chemokines, and many different cell subsets, this is a challenging task (*Davis et al., 2017*). Liquid mass cytometry (LMC), imaging mass cytometry (IMC), and flow cytometry (FC) are powerful techniques applied to exploratory immunophenotyping and biomarker discovery. With the ability to profile over 40 markers on an individual cell, these techniques have rapidly expanded our understanding of the immune system and its perturbations throughout disease pathogenesis (*Bertolo et al., 2020*; *Wang et al., 2020*). However, such comprehensive analyses produce large amounts of data and increased dimensionality, resulting in a demanding computational task to analyze (*Newell and*

*Cheng, 2016*). Often, a limiting factor to these analyses is the requirement of an in-depth knowledge of computational biology. There is an unmet need for an easy-to-use flexible open-source computational framework to explore a variety of high-dimensional single-cell cytometry datasets such as LMC, IMC, and FC within a single framework for nonspecialists. A number of previous computational cytometry workflows have been proposed, including *CyTOF workflow* (*Nowicka, 2019*), (*Diggins et al., 2015*) and *CapX* (*Ashhurst, 2019*). Subsequently, some of these workflows have been incorporated into graphical user interfaces (GUI) or code-based workflow packages such as *CATALYST* (*Crowell et al., 2020*), *Cytofkit* (*Chen et al., 2016*), *diffcyt* (*Weber et al., 2019*), and *Spectre* (*Ashhurst, 2020*). However, these approaches can be challenging to implement without significant computational knowledge. We have therefore developed *ImmunoCluster,* building on and extending the framework proposed in the *CyTOF workflow* (*Nowicka, 2019*), to create modular, flexible, and easy-to-use implementations of cytometry analysis pipelines. The *ImmunoCluster* package is a framework that provides appropriate data structures and methods to increase the utility of these high-dimensional methods and make data analysis and interpretation accessible to all researchers to facilitate immune phenotyping projects, such as those associated with preclinical and clinical studies.

Here, we present *ImmunoCluster,* an open-source computational framework for the analysis of high-dimensional LMC, IMC, and FC datasets. *ImmunoCluster* is an R package and framework that focuses on the organization and visualization of data to help define biological identity of cells, construct a hierarchy of biologically meaningful cell populations, and detect significant changes between conditions, timepoints, and groups, including serial samples from a clinical trial setting. The framework uses state-of-the-art computational techniques in an easily amendable and flexible format. Users can manipulate figures and outputs to suit their specific needs. The computational approaches used to analyze high-dimensional data are rapidly evolving, and *ImmunoCluster* has the flexibility to incorporate these novel computational methods in dimensionality reduction, clustering, and trajectory analysis within its framework. The computational framework is designed for ease of use and broad applicability so that it is simple enough to be implemented by users with only a basic knowledge of R, as well as possessing the flexibility for more advanced users to incorporate additional new methodology and build on the framework. The framework relies on the SingleCellExperiment class (*Lun, 2017*), which means that the flow cytometry standard (FCS) data is contained within a purpose-built object that stores all stages of analysis to permit multiple analysis paths to be performed in parallel. The framework incorporates methods for applying popular R packages (i.e., *RPhenograph* [*Chen and Chen, 2015*], *Rtsne* [*Krijthe, 2018*], and *FlowSOM* [*Van Gassen, 2015a*]) along with convenient 'wrapper functions' allowing for the use these popular dimensionality reduction and clustering algorithms in an easy format for nonspecialists to rapidly produce interpretable data that are extendable to advanced experimental designs. *ImmunoCluster* also contains functionality for adaptive downsampling at the import and dimensionality reduction stages of the pipeline to overcome significant increases in runtime associated with large datasets in dimensionality reduction algorithms like UMAP and tSNE. The outputs from *ImmunoCluster* are created using the *ggplot* package, which generates ggplot graphical objects that are flexible, can be further modified and utilized directly into figures for reports and publications. The core of the visualization tool was developed using *scDataviz*, a Bioconductor package (*Blighe, 2020a*) for visualizing single-cell data and influenced by the visual framework outlined in *CyTOF workflow* (*Nowicka, 2019*). *ImmunoCluster* offers an alternative to the currently used frameworks such as *CATALYST* (*Crowell et al., 2020*) and *Cytofkit* (*Chen et al., 2016*), with a focus on ease of use for biologists, as well as offering additional novel aspects for its users compared with other published frameworks. Importantly, and unique to *ImmunoCluster*, the scope of functionality in this package also enables the analysis of IMC data. Users are able to run IMC data in line with FC and LMC data analyses. Due to the lower resolution of IMC data (compared with FC and LMC), we specifically designed the 'ranked expression heatmap' function to work with IMC data, which allows users to rank markers measured to help identify cell populations. IMC provides a means to analyze the spatial dimension of the cells in situ within tissues, which can provide important insight when ascribing functionality to cell populations. Methods for analyzing large cytometry datasets, particularly IMC datasets, in an open-source computational environment are currently limited. *ImmunoCluster* has been designed for use by a nonspecialist; however, it would be useful to a range of users from wet-lab nonspecialist experimentalists to experienced computational biologists with potential utility in day-to-day research through to large-

scale studies involving multiple longitudinal serial biopsies where samples need to be analyzed and compared.

Clinical immune monitoring is an area of increasing utilization (*Hernandez et al., 2020*). In the current work, we will present examples of *ImmunoCluster's* ability to detect clinically relevant perturbations in immune cellular heterogeneity in LMC, IMC, and FC datasets in human patient and/or healthy samples. *ImmunoCluster* is easily extensible to several arbitrary experimental designs and can be utilized to aid both discovery of novel subpopulations of cells within heterogeneous samples or as a foundation for monitoring longitudinal changes or responses to treatment. As such, *ImmunoCluster* provides a resource that will permit unsupervised high-dimensional data analysis to be more widely adopted in immunobiology and many other disciplines.

## Results

This section aims to convey the scope of *ImmunoCluster*, its versatility, and applicability to complex datasets across a range of high-dimensional approaches. We implemented the *ImmunoCluster* framework for three different types of high-dimensional single-cell data (LMC, IMC, and FC). *ImmunoCluster* also offers user-friendly flexibility throughout the framework, and researchers can easily adapt all figures and outputs to suit their specific needs, resulting in an abundance of tailored outputs for the user to assess and use in publications, reports, and presentations. An example detailing a variety of tailor-made figures produced by *ImmunoCluster* can be found in *Figure 1—figure supplement 1*.

We first used previously published and publicly available LMC data (*Hartmann et al., 2019*). This dataset was chosen as it is representative of an intricate immunophenotyping project and would test the ability of *ImmunoCluster* to reproduce published results. Secondly, we investigated two novel IMC datasets, from HNSCC and diffuse large B-cell lymphoma (DLBCL) patients with the aim of demonstrating the applicability of the *ImmunoCluster* tool for IMC data analysis and cell cluster identification and visualization. Finally, FC data from the BM of seven healthy donors (HDs) during hip surgery were used to show the ability of the framework to analyze conventional FC data and identify rare immune cell populations.

### Liquid mass cytometry

The chosen dataset (*Hartmann et al., 2019*) includes mass cytometry data from 15 patients with leukemia who all received bone marrow transplant (BMT), three of whom developed acute graft versus host disease (GvHD). Samples were taken at 30 and 90 days post-BMT. Thirty-three cell markers, both lineage and functional, were analyzed using mass cytometry (Helios CyTOF system). Prior to uploading into the *ImmunoCluster* framework, the FCS files were normalized and gated as described in the Materials and methods section. We observed that a typical pipeline run on the full 2.3 million cell LMC GvHD dataset (with UMAP downsampling to 500k cells) would take ~110 min on a 2.9 GHz Intel Core i7 MacBook pro with 16 GB RAM (*Figure 1—figure supplement 2*).

An initial visualization of the data was first carried out using multidimensional scaling (MDS) plots, which were overlaid with metadata to identify the similarity of patients by condition and day of measurement (*Figure 1A, B*). A heatmap was also created to give an overview of marker expression for each patient and annotated with the metadata provided (*Figure 1C*). The *UMAP* algorithm was used for dimensionality reduction of data. There was a difference in the distribution of cells from GvHD and 'none' across the cell islands between both timepoints, 30 and 90 days after BMT (*Figure 2A*).

The mass cytometry panel used in this experiment was specifically designed to identify all major human immune cell lineages (*Hartmann et al., 2019*); the marker expression across cell islands was visualized and indicated cell types present within the cell islands (*Figure 2B*) (expression of all markers measured; *Figure 2—figure supplement 1*). To further distinguish the cell types that were present in the cell islands and visible from the dimensionality reduced *UMAP* plots (*Figure 2B*), the *FlowSOM* clustering algorithm with consensus clustering metaclustering was applied, which has been demonstrated to scale well to large cytometry datasets (*Weber and Robinson, 2016*). The number of clusters input into the *FlowSOM* algorithm was slightly higher than the number of expected cell clusters (n = 56) (*Figure 2—figure supplement 2*). The SingleCellExperiment object stored the data for each *K meta*-clustering (*K1-K56*); therefore, further downstream data exploration was carried out looking at different numbers of *K* clusters. A heatmap showing the expression of the

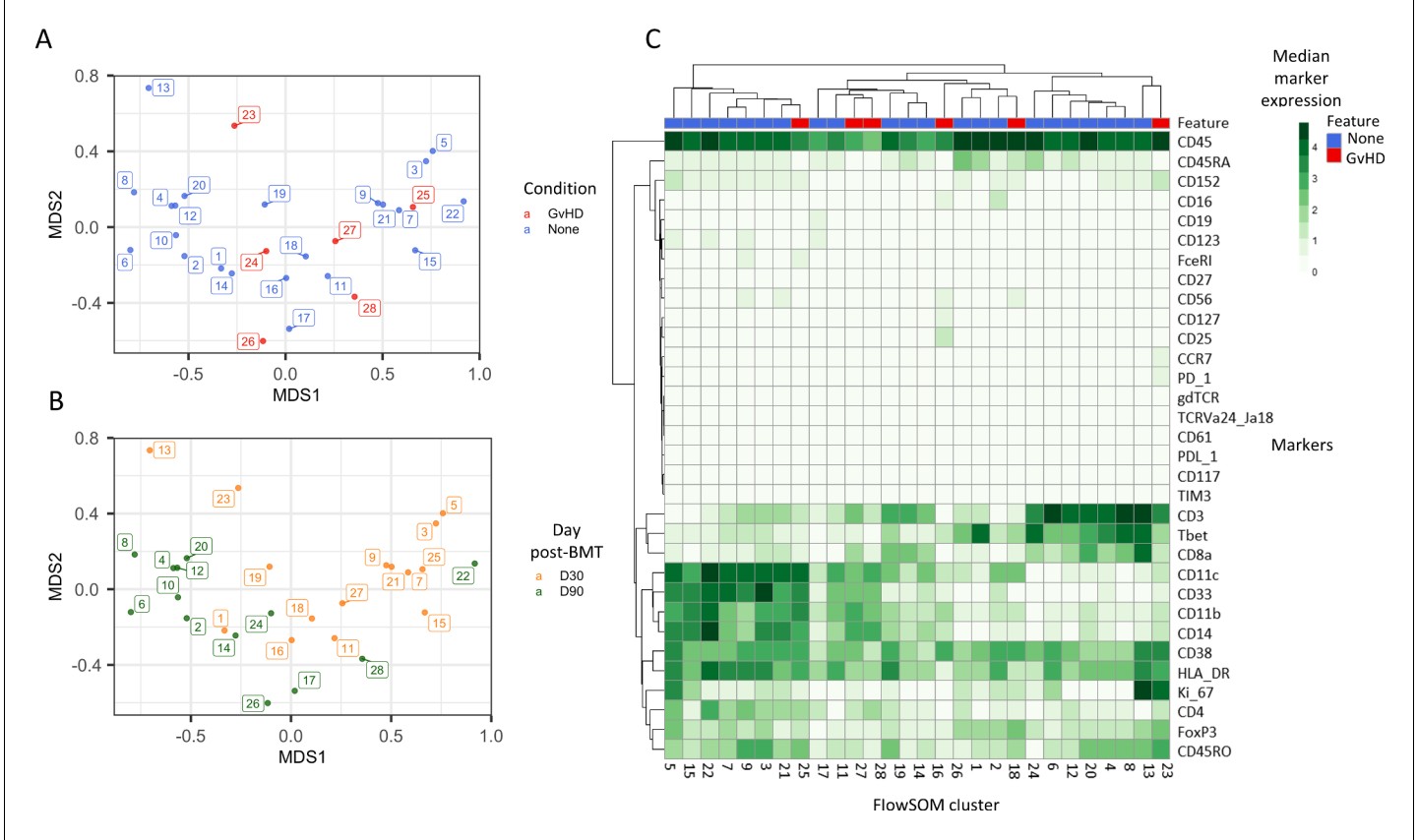

**Figure 1.** Initial exploration of liquid mass cytometry data from patients with leukemia who received bone marrow transplants (BMTs). (A, B) Multidimensional scaling of data; median marker expression data from each sample were used to create plots, annotated with condition (graft versus host disease or none) (A), and time after BMT treatment (B). (C) Heatmap showing the median marker expression for each patient. The online version of this article includes the following figure supplement(s) for figure 1:

**Figure supplement 1.** Customizable figure outputs.

**Figure supplement 2.** *ImmunoCluster* typical pipeline runtime.

33 markers measured was used to identify cluster cell types (*Figure 2—figure supplement 2*), and *ImmunoCluster* was successfully able to replicate the findings from *Hartmann et al., 2019*, with 24 of the cell populations being identified (*Figure 2C, D*). Selecting the correct number of clusters to input into the clustering algorithm can be a challenging aspect of these analyses, but the framework provides the user with multiple parameters to aid this decision. We recommend that users should always overestimate the number of cell populations as data for each *K* number of clusters will be stored in the SingleCellExperiment object and can therefore be explored and refined later using the different visualization techniques available within the framework. Additionally, the elbow plot generated by the *ConsensusClusterPlus* package can be examined alongside the clustering algorithm output to guide the decision on the best number of clusters for downstream analysis (*Figure 8—figure supplement 6*). Over-clustering can allow for the identification of rare cell types within a dataset at the expense of often generating several clusters of highly prevalent cell types that likely represent the same biological cell type. *ImmunoCluster* provides the tools to manually and reproducibly merge clusters of the same biological cell identity into one group after over-clustering. These clusters can additionally be merged into each other to use a less granular, higher-level population annotation (see *Figure 2—figure supplement 3* for an example of higher-level clustering).

The abundance of all cell types across all samples was measured, and CD14$^+$CD16$^-$ monocytes were identified as the most abundant population, correlating with the Hartmann et al. data (*Hartmann et al., 2019*; *Figure 2E*). Individually displaying patient's cell-type abundance means that

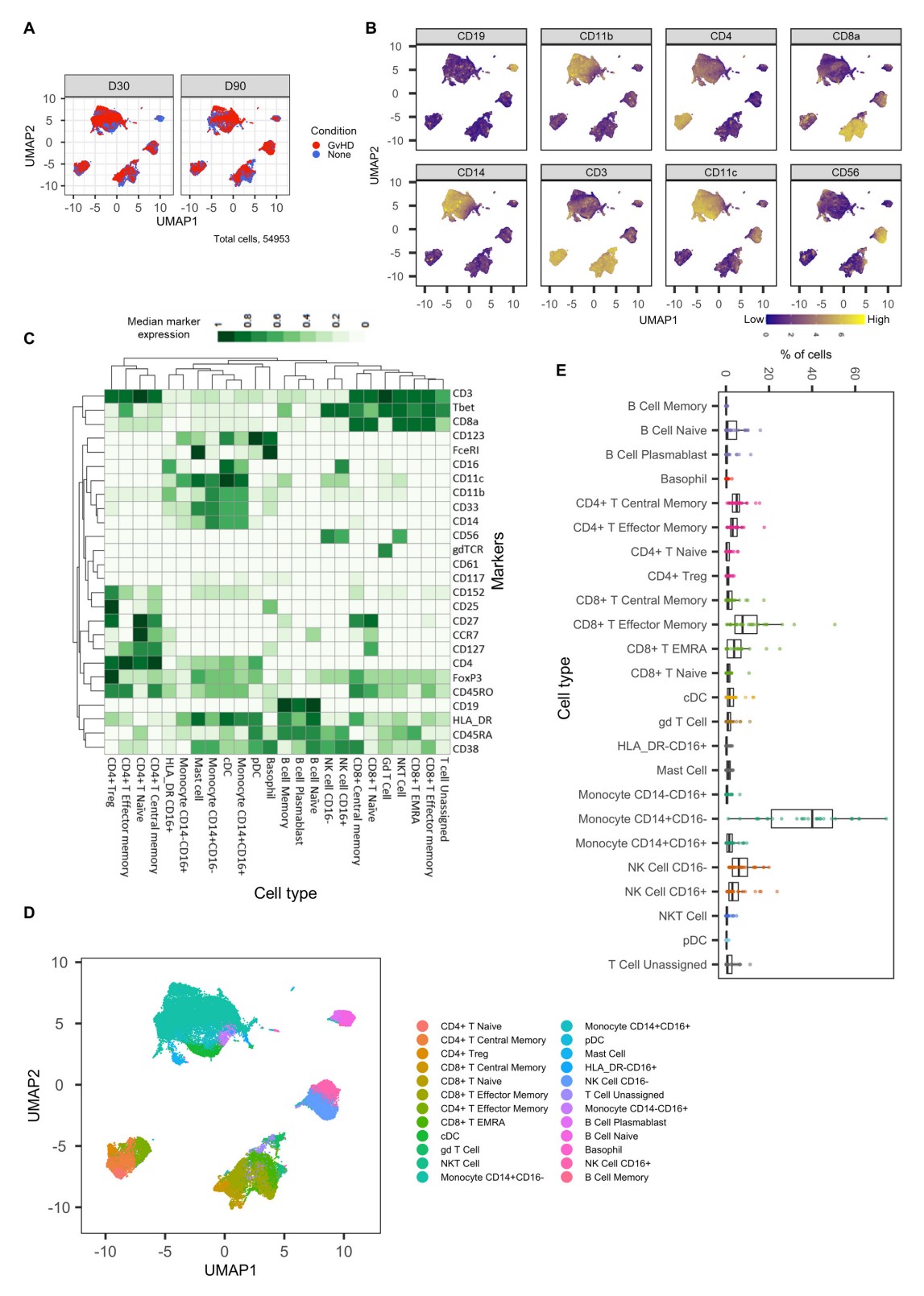

**Figure 2.** Dimensionality reduced liquid mass cytometry CyTOF data and marker expression. (A) *UMAP* plots colored by graft versus host disease and none and split by timepoint (30 and 90 days after bone marrow transplant treatment). (B) Expression of eight selected lineage markers projected onto *UMAP* plot. Identifying cell types and abundance of clusters. (C) Heatmap showing median marker expression across all identified cell types. (D) *UMAP* annotated with cell types. (E) Distribution of immune cell frequencies and abundance (%) of each cell type across all samples measured.
*Figure 2 continued on next page*

we could identify variation within a group of interest; for example, we observed that although there is variation within the GvHD and none, overall, they appear to follow the same trends in cell-type abundance (*Figure 3A*). Additionally, significant differences between memory B-cells (false discovery rate [FDR] p=4.38×10$^{-3}$), naïve B-cells (FDR p=1.35×10$^{-2}$), and naïve CD4$^+$ T-cells (FDR p=3.47×10$^{-2}$) were identified between the GvHD and none (*Figure 3B, C*) using a t-test. A difference in number of naïve B-cells was previously reported by *Hartmann et al., 2019* in this dataset and can be seen in *Figure 2A* (using *Figure 2D* for cell island identification), where there is a noticeable reduction of these cells in the GvHD patients. In addition, a reduction in naïve CD4$^+$ T-cells can also be seen, which was also previously reported, but to a lesser extent. A volcano plot can be used to highlight the differentially abundant cell clusters between GvHD and none (GvHD logFC+ve and none logFC-ve), where cell types with FDR p<0.05 are shown in red (*Figure 3D*), which is also in line

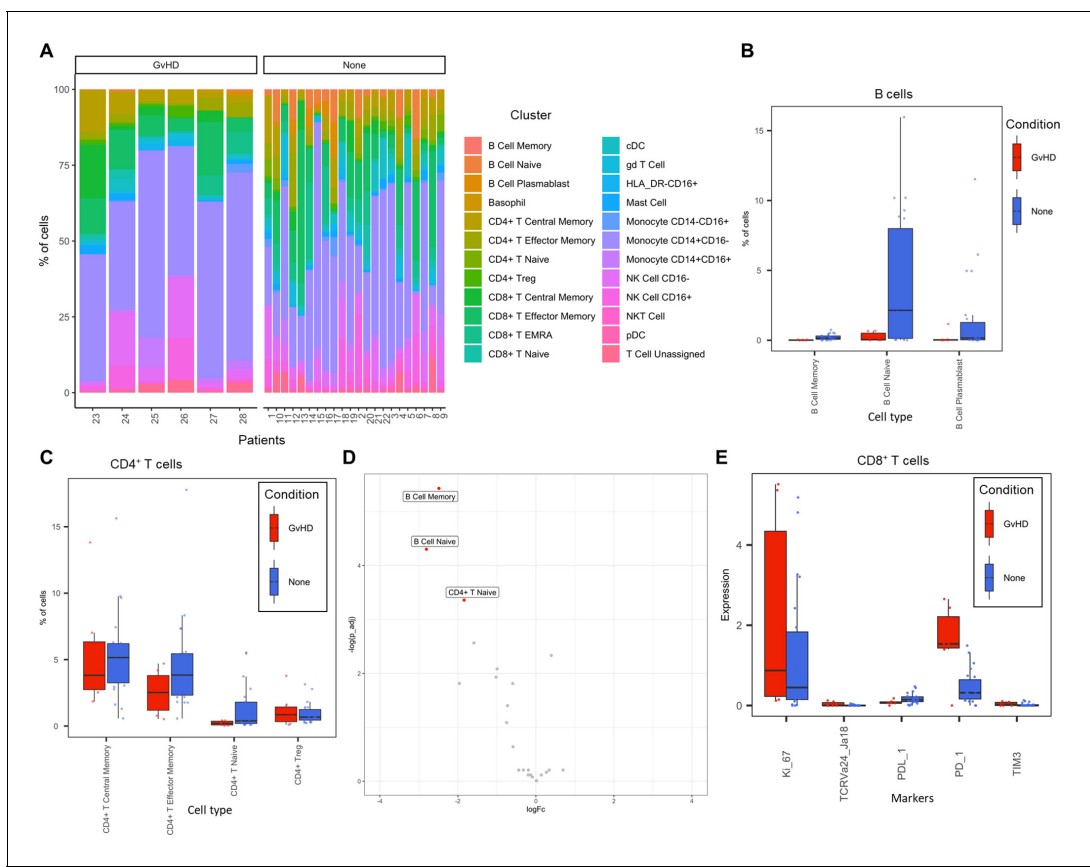

**Figure 3.** Liquid mass cytometry (LMC) CyTOF data cluster and cell-type abundances. (**A**) Percentage of each cell type shown for each patient numbered across the bottom of the plot. (**B, C**) Box plots portray interquartile range (IQR) with the horizontal line representing the median percentage of cell types in both graft versus host disease (GvHD) and none patients for B-cell and CD4$^+$ T-cell populations, respectively. (**D**) Differential abundance analysis; volcano plot showing the significantly differentially expressed cell abundances (false discovery rate p<0.05) between GvHD and none (GvHD logFC+ve and none logFC-ve). (**E**) Comparison of checkpoint-related molecules (PD-1 and TIM3), receptors (PD-L1), proliferative (Ki-67), and iNK T-cells (TCRVa24-Ja18) marker expression between GvHD and none patients in the CD8$^+$ T-cell cluster.

The online version of this article includes the following figure supplement(s) for figure 3:

**Figure supplement 1.** Biaxial plots.

with the previously published data (*Hartmann et al., 2019*). We compared *ImmunoCluster's* differential abundance testing output (*stat_test_clust*), run in the t-test mode, with the Hartmann et al. data (*Hartmann et al., 2019*) into the *diffcyt* computational framework (*Weber et al., 2019*), which is a state-of-the-art tool for differential discovery analyses. *Diffcyt* identified the same three cell clusters (memory B-cells, naïve B-cells, and naïve CD4[+] T-cells) as differentially abundant (FDR p<0.05) between the GvHD and none conditions. This demonstrates concordance with *ImmunoCluster's* statistical output that identified naïve B-cells and naïve CD4[+] T-cells as one of the principal differentially abundant populations, which was also identified in the original analysis of the data (*Supplementary file 4*). Additionally, we explored the expression of checkpoint-related molecules, and their receptors, markers of proliferation, and invariant natural killer T (iNK T) cells in CD8[+] T-cells and compared the expression of these in the patients with GvHD and none (*Figure 3E*). These markers help assess the functional states of cells and the checkpoint-related molecules and proliferative activity markers such as PD-1, PD-L1, TIM3, Ki-67, and TCRVa24-Ja18; the expression of these antigens has been proposed previously as candidate biomarkers for immunotherapy (*Hartmann et al., 2019*). Visually, GvHD patients had a higher expression of PD-1 and a noticeable difference in the Ki-67 proliferation marker as expected for these patients (*Figure 3E*).

## Imaging mass cytometry

We utilized two different IMC datasets to demonstrate how *ImmunoCluster* would handle IMC data from two different tumor microenvironments that possess (1) a biological image with clear boundaries, such as head and neck squamous cell carcinoma (HNSCC), where clear tumor and stroma regions were evident; (2) a biological image that was more heterogeneous and diffuse, such as the DLBCL lymph node section. *ImmunoCluster's* ability to explore IMC data adds an important element to the framework and demonstrates the flexibility and applicability of the tool. Researchers can explore IMC data easily, and in line with LMC and FC data. After data preprocessing (see Materials and methods and *Figure 8—figure supplement 2*), the transformed gated FCS files were uploaded into the *ImmunoCluster* framework. Generally, the same workflow that was applied to the LMC and FC data was applied to the IMC data from a section of tissue from patients with HNSCC and DLBCL (lymph node) as a proof-of-principle application of the *ImmunoCluster* tool for IMC data analysis (*Figures 4* and *5*). The only difference with the IMC data was the use of a rank heatmap (*Figure 4—figure supplement 1*); this ranks the order of each marker for each sample (1 – total number of samples). The reason for this was clarity and ease of cell identification across clusters as IMC produces lower resolution data compared to the data output of LMC or FC, which makes manual assignments of populations more difficult to interpret. Therefore, a rank of expression was used as an easy means to facilitate the assignment of cell identity through defining the highest and lowest expression of markers between clusters.

## Head and neck squamous cell carcinoma tissue section

Cell segmented regions are annotated in *Figure 4A*, where regions 1–3 were selected as tumor regions and 3–6 as stroma regions. The *ImmunoCluster* tool was able to distinguish between the stroma and tumor regions of the tissue section from the HNSCC tumor (*Figure 4B*), and with additional markers could provide a more in-depth analysis of the cell phenotypes in each respective region of the tumor. Dimensionality reduction of the data clearly separated the cell islands belonging to the stroma and tumor regions of the tissue (*Figure 4C*), suggesting different cell phenotypes between the regions. The *FlowSOM* algorithm was applied to cluster the cells and identify cell populations. The rank heatmap was used to show the ranked expression of each marker (ranked 1–10, with 10 being high) for each cluster (1–10) (*Figure 4—figure supplement 1*), and the identified cell types were then used to annotate the heatmap (*Figure 4D*). High expression of E-cadherin can be used as a marker of cancerous tissue in HNSCC, and its expression correlates with the cancer regions selected for analysis (*Figure 4D*). We focused on CD8[+] T-cell, B-cell, and macrophage cell clusters (*Hladíková et al., 2019*; *Evrard et al., 2019*). Regions 2 and 3 of the tumor mostly consist of proliferating (Ki-67[+]) tumor cells and macrophages (tumor-associated and PD-L1[+]). Region 1 was a mix of immune cells, B-cells, and PD-L1[+] CD8[+] T-cells (*Figure 4E*). Stromal regions 5 and 6 are mostly PD-L1[+] macrophages and CD8[+] T-cells. Region 4 has a cluster of mixed immune cells, and from the IMC image (*Figure 4A*) we can see that this region looks like the tumor is encroaching into the stromal

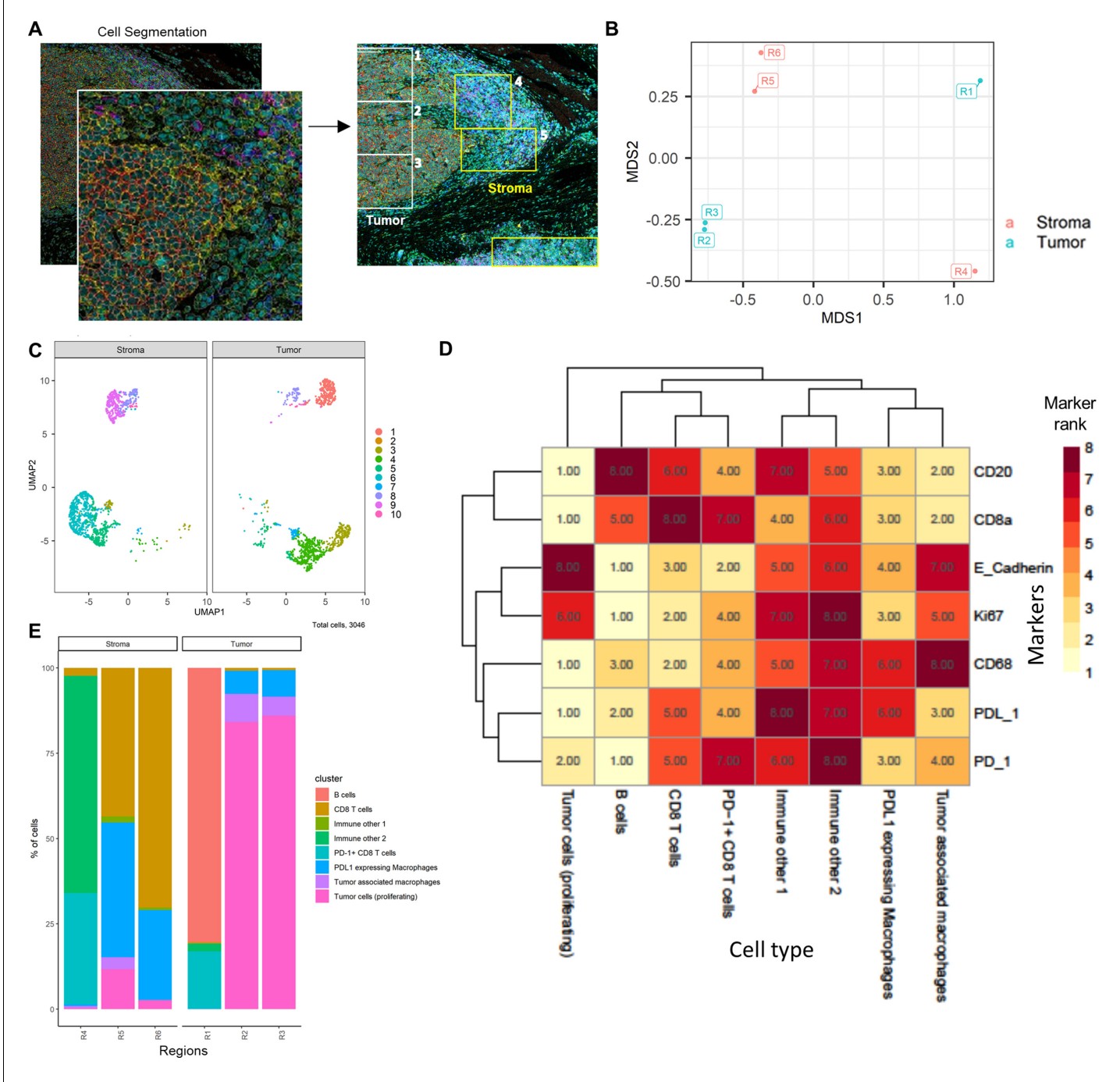

**Figure 4.** HNSCC imaging mass cytometry (IMC) data. Immunophenotyping the tumor microenvironment with IMC data using *ImmunoCluster*. (**A**) IMC image showing an example of five channels: PD-L1 (green), CD4 (yellow), E-cadherin (red), CD20 (magenta), and CD8α (blue). Images with the segmented cell borders are highlighted. The tumor and stroma areas are clear to the eye, and three regions were selected from each as shown (regions 1–6). (**B**) Multidimensional scaling plot of stroma and tumor regions. (**C**) Dimensionality reduced data (*UMAP* algorithm applied) annotated with *FlowSOM* clusters and split by region type. (**D**) Rank heatmap: ranked expression (1–8, where 8 is high) of seven markers (CD20, CD8α, E-cadherin, Ki-67, PD-L1, CD68, and PD-1) and identified cell type. (**E**) Proportion of cell types for each tissue region.

The online version of this article includes the following figure supplement(s) for figure 4:

**Figure supplement 1.** Rank heatmap for the head and neck squamous cell carcinoma patient imaging mass cytometry data.

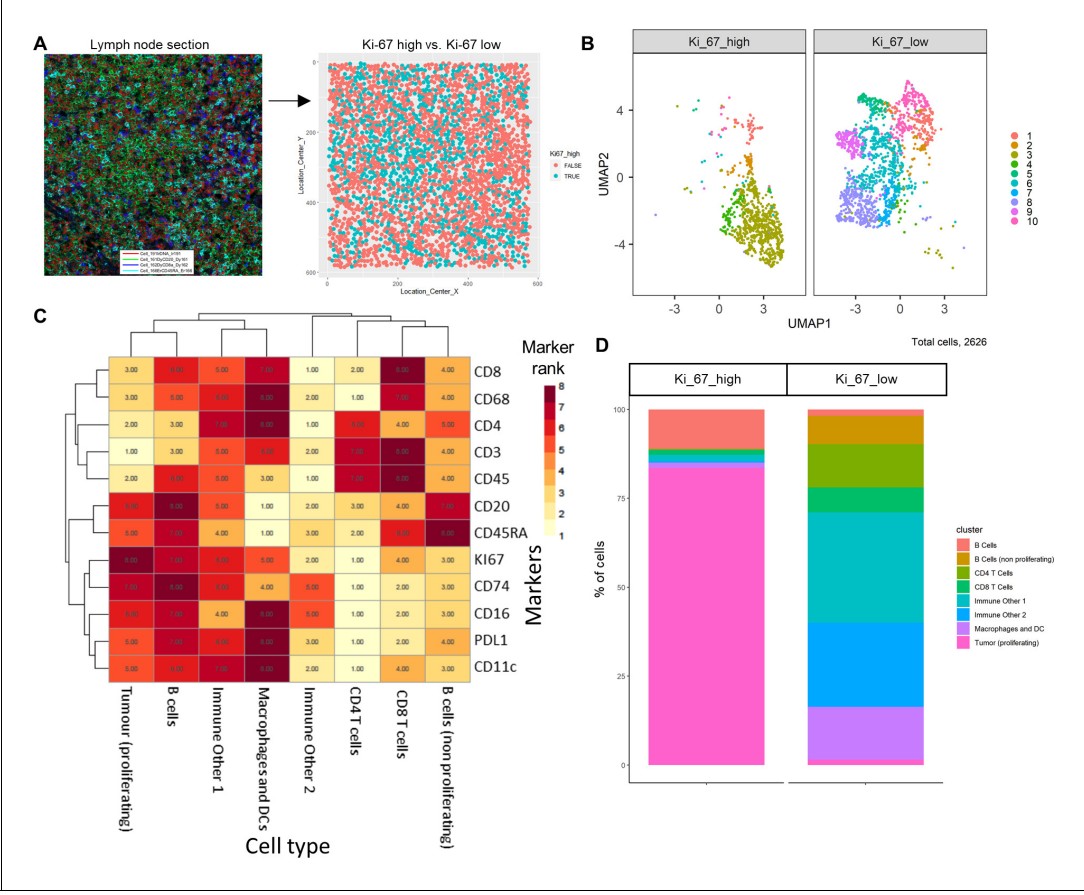

**Figure 5.** Diffuse large B-cell lymphoma imaging mass cytometry (IMC) data. Immunophenotyping the lymph node microenvironment with IMC data using *ImmunoCluster*. (**A**) IMC image showing an example of four channels: DNA (red), CD20 (green), CD8α (dark blue), and CD45RA (light blue). The image was split by high and low Ki-67 expression to identify highly proliferative cells (tumor). (**B**) Dimensionality reduced data (*UMAP* algorithm applied) annotated with *FlowSOM* clusters and split by Ki-67 high or low. (**C**) Rank heatmap: ranked expression (1–8, where 8 is high) of 12 markers (CD8, CD68, CD4, CD3, CD45, CD20, CD45RA, Ki-67, CD74, CD16, PD-L1, and CD11c) and identified cell type. (**D**) Proportion of cell types for Ki-67 high and Ki-67 low cell populations.

The online version of this article includes the following figure supplement(s) for figure 5:

**Figure supplement 1.** Rank heatmap for the diffuse large B-cell lymphoma patient imaging mass cytometry data.

region. The stroma seems to be mostly nonproliferating cells (Ki-67⁻ compared to the tumor region) and CD8⁺ T-cells, including a CD20⁺ CD8⁺ cytotoxic subset (*Gingele et al., 2020*).

## Diffuse large B-cell lymphoma lymph node section

The IMC DLBCL lymph node section was split by high and low Ki-67 expression to identify the highly proliferative tumor cells (*Figure 5A*); Ki-67 is used as a prognostic marker in DLBCL (*Tang et al., 2017*). The *FlowSOM* algorithm was applied to the data to split the IMC data into clusters; these clusters were projected onto the *UMAP* plot to visualize how these clusters were split between the Ki-67 high and low (*Figure 5B*). The marker expression ranking tool was applied to the clustered data and used to identify cell types (*Figure 5C, Figure 5—figure supplement 1*). The majority of Ki-67 high population were proliferating tumor cells (84%), and the Ki-67 low population consisted of a heterogeneous collection of immune cell populations, such as CD4⁺ T-cells, CD8⁺ T-cells, B-cells, macrophages, and dendritic cells (DCs) that were successfully identified using the *ImmunoCluster* tool (*Figure 5D*).

## Flow cytometry

The HD BM taken during hip surgery was gated for the CD3$^+$CD4$^+$ population before analysis within the *ImmunoCluster* framework, with the aim of testing *ImmunoCluster's* ability to identify minor populations such as Tregs and their subpopulations (*Kordasti et al., 2016*). The *FlowSOM* algorithm was applied to the data, resulting in 40 clusters (*Figure 6—figure supplement 1*), and a heatmap was created with median expression of markers to identify cell types (*Figure 6B*). Due to the markers used and the low prevalence of Tregs, the majority of cells were identified as CD4$^+$ T-cells. Additionally, three populations of Tregs were identified by the *ImmunoCluster* tool: Tregs (CD25$^+$CD127$^{low}$) (3.6%, 1.3–5.5), Treg A (CD25$^+$CD127$^{low}$ CD45RA$^+$) (0.7%, 0.1–2.0), and Treg B-cells (CD25$^+$-CD127$^{low}$CCR4$^+$CD95$^{high}$) (0.9%, 0.2–2.4) (*Figure 6C*). The abundance of Treg A and B for all HDs is shown in *Figure 6D*, and their distribution was as expected (*Kordasti et al., 2016*).

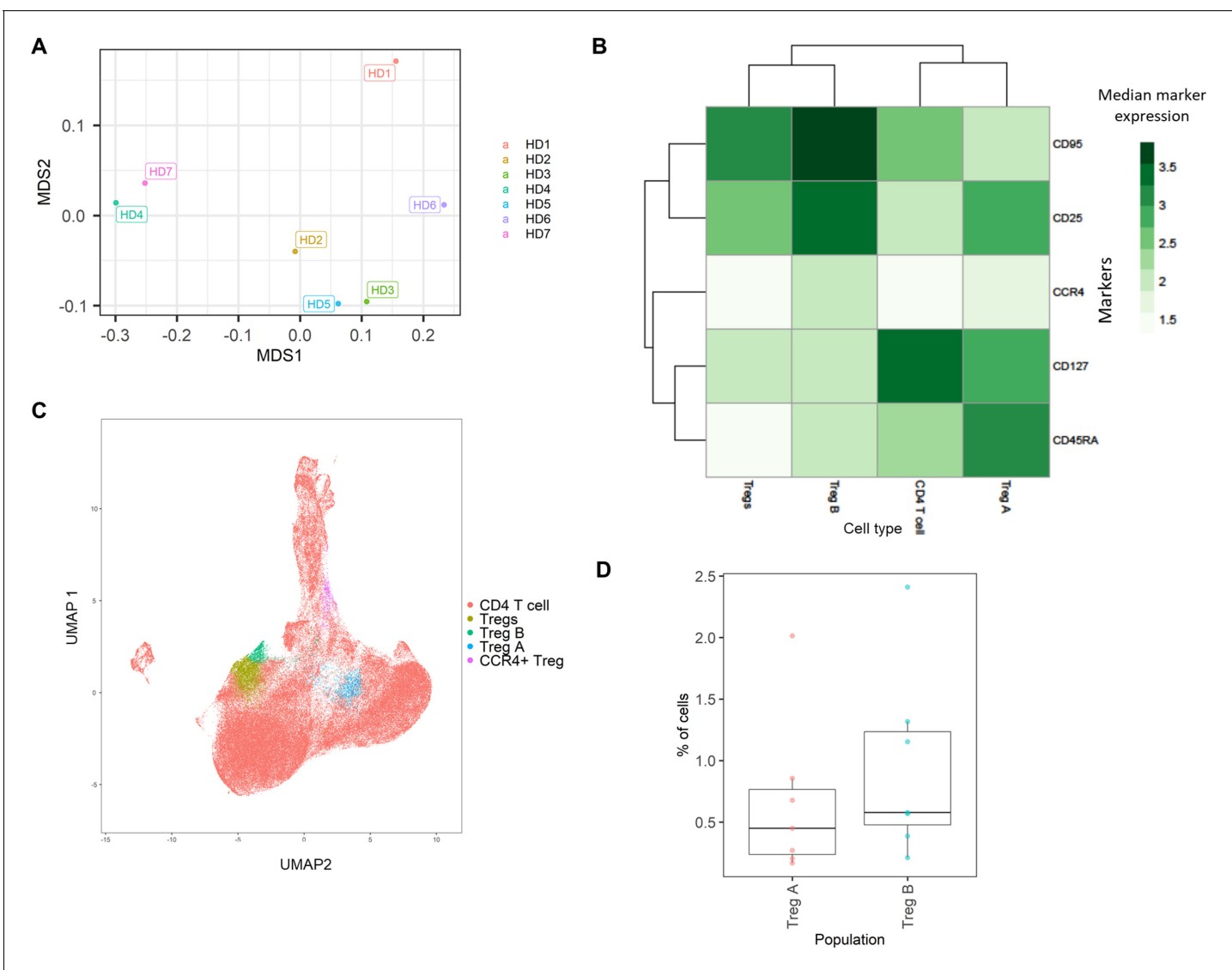

**Figure 6.** Healthy donor bone marrow flow cytometry data: identification of rare CD4$^+$ T-cell immune cell subsets. (**A**) Multidimensional scaling plot of each healthy donor (1–7). (**B**) Heatmap showing marker expression across clusters of identified cell types. (**C**) Dimensionality reduced data (*UMAP* algorithm applied) annotated with cell type. (**D**) Proportion of Treg A and B from total cells from healthy donors.

The online version of this article includes the following figure supplement(s) for figure 6:

**Figure supplement 1.** Heatmap showing the marker expression of 40 *FlowSOM* clusters.

## Discussion

Here, we provide an overview of the flexibility and scope for utilizing *ImmunoCluster*, an open-access easy-to-use framework for LMC, IMC, and FC dataset analyses. The *ImmunoCluster* package emphasizes ease of use in the generation of flexible and modular cytometry analysis pipelines for applications such as clinical immune monitoring studies. Explanations of the functions and workflow have been provided, and an in-depth step-by-step protocol and tutorial are available on the GitHub site (https://github.com/kordastilab/ImmunoCluster).

The purpose for creating the *ImmunoCluster* framework was to support nonspecialist researchers to carry out complex immunophenotyping experiments. The dawn of 'cytometry big data' and its ever-growing utilization has surpassed the number of experimental parameters that a researcher can feasibly analyze as a collective. The *ImmunoCluster* framework was designed in collaboration between wet lab and computational biologists, with the purpose of creating a tool that enables state-of-the-art analysis, yet easy to use in its application. We provide sufficient detail and explanation for the nonspecialist researcher to understand the need for each analysis step as well as how to confidently execute the process, with an aim of performing transparent and reproducible analyses.

*ImmunoCluster* has been built with an emphasis on generalizability and scalability to facilitate a broad use and is an advance over other open-source tools as it provides an integrated framework to uniquely perform a complete computational pipeline on high-dimensional LMC, IMC, and FC data. The inclusion of IMC data analysis brings a unique element to the *ImmunoCluster* framework and differentiates it from other published protocols. *ImmunoCluster* was built to easily leverage many popular R packages for cytometry data (i.e., *RPhenograph* [*Levine et al., 2015*], *Rtsne* [*van der Maaten and Hinton, 2008*; *Laurens van der, 2014*], and *FlowSOM* [*Van Gassen, 2015b*]), selected because they are open-source and regularly maintained with extensive documentation. All methods prioritize customizable and attractive visualizations designed to be used by both dry- and wet-lab researchers/clinicians. The inclusion of adaptive downsampling of cells for the running of computationally intensive dimensionality reduction steps, using tools such as *UMAP* and *tSNE*, allows *ImmunoCluster* to generate cytometry analysis pipelines that are readily scalable to a dataset of millions of cells across several samples and a variety of experimental or phenotypic conditions. *ImmunoCluster* can also be run on a local desktop/laptop computer with standard configuration, depending on the number of cells, adaptive downsampling settings and selection of clustering algorithms can usually be run within a day end-to-end (*Figure 1—figure supplement 2*). Whilst the *PhenoGraph* clustering method does not readily scale, the *FlowSOM*-based clustering method can be implemented on much larger datasets.

The limitations of this computational framework are important to take into consideration when designing an experiment. Unsupervised clustering (stage 2) of samples is the most important stage of the framework, and its ability to accurately define populations across all samples is critical to all later stages of investigation. The number of *k* clusters to generate (cluster resolution) can significantly affect the ability to identify or determine a change in a biologically relevant population. Merging two subpopulations due to low resolution of clustering may mask an important experimental observation. Determining the precise number of clusters that are relevant for a given dataset is an important step. In our step-by-step guide, we provide complete documentation and clear user-friendly approaches to optimally define this.

Technical inter-sample marker signal variability due to batch effects may impact the ability of unsupervised analysis to reliably detect certain populations if significant batch effects are present. MDS plots created in stage 1 of the framework can be used to detect batch effects as well as other technical artifacts such as antibody staining anomalies. An approach commonly employed in mass cytometry to overcome this is to use sample barcoding, reducing the variability between each sample, allowing all samples to be exposed to the same antibody mixture (*Schulz and Mei, 2019*). Another regularly employed approach is the inclusion of a shared control sample in each independent batch, for example, the same cell type as the experimental samples but all from the same HD. Statistical methods for batch correction may also be applied; in recent years, a class of methods called remove unwanted variation (RUV) have been developed, and *CytofRUV* (*Trussart et al., 2020*) is a recently developed package specifically designed for CyTOF dataset batch correction. If there are significant differences in the total number of cells recovered between samples, samples with many more cells may bias the clustering. Samples with very few cells recovered may result in

information loss and missing populations that are in fact present. The *ImmunoCluster* framework provides users with two opportunities (creating the SingeCellExperiment and running the dimensionality reduction) to downsample, which means the same number of samples will be used for each sample, rectifying the problem of varying cell numbers. If a particular sample has very few cells, they may need to be excluded from the analysis as they may not be a representative sample. Although channel spillover is diminished in mass cytometry, it still exists in fluorescence-based FC and should be considered when designing antibody panels to reduce the effects on introducing cell phenotype artifacts in downstream unsupervised analysis. As such, initial exploratory data analysis (e.g., MDS plots) is key to determining if any of these confounding factors might be present in the data. Additionally, users may want to introduce fluorescence-minus-one controls, where cells are stained with all fluorescently tagged antibodies except for the one of interest (*Roederer, 2001*).

## Summary

The *ImmunoCluster* package increases the accessibility of advanced computational methods for users tasked with generating high-quality analysis of high-dimensional LMC, IMC, and FC data. More advanced users can leverage *ImmunoCluster*'s data structures, methods, and suitability for scripted analysis to integrate our work with other R analysis tools and build on the package's foundation. In our clinically relevant immune monitoring case study setting, the *ImmunoCluster* framework successfully identified 24 phenotypically distinct cell clusters, and their abundance across all samples, and highlighted differentially represented cells between GvHD and none patients, all of which were consistent with the Hartmann et al. data (*Hartmann et al., 2019*). We also applied the framework to two sets of IMC data, showing the broad applicability of *ImmunoCluster* as well as the ease of being able to analyze the IMC data in line with LMC data, helping compare these types of data, which will be useful in future studies where multiple technologies are being applied within one study. Finally, we implemented FC data into *ImmunoCluster*, confirming that it was able to analyze conventional FC data and identify rare immune cell populations. Currently, the preprocessing of raw files is carried out prior to implementation into the *ImmunoCluster* framework, but due to the flexibility and design of the tool, future implementations may include IMC data preprocessing within the *ImmunoCluster* R package. This versatile framework provides an opportunity to include all researchers with varying knowledge and experience in computational biology to be involved in the experimental project from experimental design, wet lab/clinical trial, all the way through the data analysis process and visualization.

## Materials and methods

### Data file preprocessing and preparation for implementation into the *ImmunoCluster* framework

LMC data

Here, we describe an example of the *ImmunoCluster* package analyzing previously published LMC data (CyTOF) from 15 patients with leukemia 30 and 90 days after BMT, previously published by *Hartmann et al., 2019*. After BMT, three of these patients suffered acute GvHD and 12 had no evidence of GvHD (noted as 'none' herein). For systems-level biomarker discovery within this trial, a panel of 33 antibodies were incorporated into the immunophenotyping panel (*Supplementary file 1*). The panel was designed to cover all major immune cell lineages and several functional subsets including T-, B-, NK-, and myeloid cells and granulocytes (detailed in *Hartmann et al., 2019*). The marker panel also included a variety of immune-regulatory proteins, such as PD-1, PD-L1, and TIM3. The publicly available dataset from the Hartmann et al. study was extracted for the current study from the FlowRepository (http://flowrepository.org/id/FR-FCM-Z244) (*Spidlen et al., 2012*). Raw FCS files for the LMC dataset that were collected on the Helios CyTOF system (Fluidigm, UK) machine were first normalized using the free CyTOF 6.7 system control software (files can also be concatenated using the same software if necessary). These files were gated using FlowJo (version 10.5.3) (Becton, Dickinson and Company, UK) to remove beads, doublets, dead cells, and non-CD45$^+$ cells as well as erythrocytes (CD235αβ/CD61$^+$) and neutrophils (CD16$^+$) (*Figure 8—figure supplement 1*), as described by *Hartmann et al., 2019*, before implementation into the *ImmunoCluster* framework. There are various alternative open-access tools that can be used to gate these

files, such as OpenCyto (*Finak et al., 2014*), CytoExploreR (*Hammill, 2020*), and Cytoverse (https://cytoverse.org/).

## IMC data

Two different examples of IMC datasets were implemented into the framework. One IMC dataset was from a human tissue section taken from a patient with HNSCC. Consent was attained by the Guy's and St Thomas' Research Biobank, within King's Health Partners Integrated Cancer Centre. The second IMC dataset was from a lymph node section from a patient with DLBCL. Formalin-fixed paraffin-embedded (FFPE) DLBCL tumor tissue was obtained from King's College Hospital, in accordance with the Declaration of Helsinki and approved by the UK National Research Ethics Committee (reference 13/NW/0040). A detailed protocol is available for both datasets as a Supplementary file (Appendix 1 and 2, respectively). The raw HNSCC IMC data from 12 channels (10 markers + 2 intercalator-Ir channels; *Supplementary file 2*) and the raw DLBCL IMC data from 20 channels (19 markers + nuclei channel; *Supplementary file 3*) were processed with Python scripts and *CellProfiler* pipelines according to *Schulz et al., 2018* to segment the image data into individual cells (*Figure 8—figure supplement 2*). The mean marker intensity for each segmented cell was exported (comma-separated values [CSV] file) and either divided into six regions of equal area (HNSCC, three tumor and three stroma regions) or divided by Ki-67 expression (DLBCL, Ki-67 high or low). The mean marker intensity for each segmented cell was multiplied by 65,535 to recover the image intensities (Fluidigm IMC machine = 16 bit, the dynamic range of the measurements is 0 to $(2^{16})-1$) and asinh (co-factor 0.8) transformed in R Studio. These CSV files were converted into FCS files and gated in Cytobank (*Chen and Kotecha, 2014*); functions are also available within the *ImmunoCluster* package to convert SCE or CSV data files into FCS format.

## FC data

FC data from the BM of seven HDs were used to demonstrate the ability of the framework to analyze conventional FC data and identify rare immune cell populations. BM samples from HDs were obtained by extraction of the bone marrow cells from the bone of the femoral head. This noninterventional study was approved by the ethical committee of Cochin-Port Royal Hospital (Paris, France) (CLEP Decision No.: AAA-2020-08039). Femoral heads were obtained after informed consent during hip replacement surgery. These were cut in half and collected in a conservation medium (Hanks balanced salt solution with $NaHCO_3$, Eurobio), supplemented with heparin (7%) and then transported to the laboratory at room temperature (RT). These were scraped with a spatula, ground in a mortar, and washed with a PBS solution supplemented with 100 µg/ml DNAse (Sigma Aldrich). The FCS files were gated for singlets using a FSCint/FSCpeak dot plot, and dead cells were removed using a forward scatter (FSC) and side scatter (SSC) dot plot. Leukocytes were gated on a CD45/SSC dot plot. Finally, they were gated to identify the $CD3^+CD4^+$ population of cells for analysis in the *ImmunoCluster* framework (gated in Cytobank; *Chen and Kotecha, 2014*).

## Workflow overview

The *ImmunoCluster* package is accessible via GitHub (https://github.com/kordastilab/ImmunoCluster), where a detailed step-by-step walkthrough of the tool and R Markdown files (RMD) for LMC/FC and IMC data analysis are available. The RMD files permit users to logically work through the framework in sections in a 'click and play' manner. Additionally, all scripts to replicate figures published here are available via GitHub. Users will need to have R downloaded to run the *ImmunoCluster* tool. The *ImmunoCluster* framework provides tools and support, allowing researchers to follow a workflow that guides them through experimental design, data analyses, interpretation, and statistical significance testing, to publishable graphics identifying differences in phenotype and abundance of cells between conditions analyzed (*Figure 2* and *Figure 7* and *Figure 8*). The framework comprises three core computational stages that are conducted by the *ImmunoCluster* tool:

## Stage 1: Data import and quality control

a. We provide tools designed primarily for computational preprocessing of FCS data files for downstream applications including parameter renaming and transformation. The generated

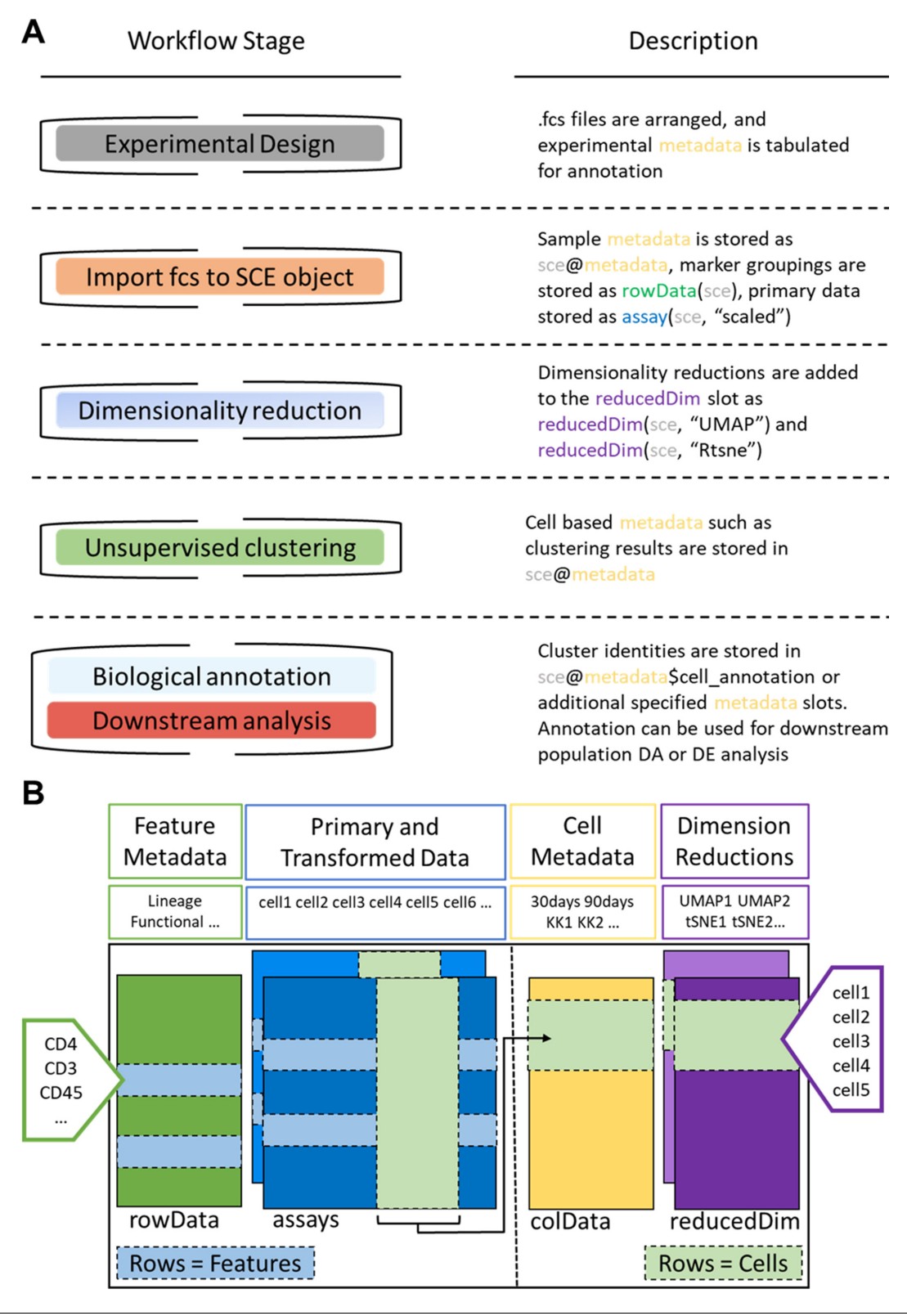

**Figure 7.** *ImmunoCluster* workflow stages and SingleCellExperiment structure. (**A**) Schematic outlining the typical cytometry workflow and its interactions with the SingleCellExperiement *ImmunoCluster* object. (**B**) SingleCellExperiment structure in *ImmunoCluster*. The SingleCellExperiment class is a data container, storing multiple layers of data to create the SingleCellExperiment object, which holds all relevant data for an experiment. Feature metadata: imported by the user in the form of a *panel_metadata* file, which is a table containing all markers measured, each annotated with

*Figure 7 continued on next page*

*Figure 7 continued*

either lineage or functional information for downstream analysis. Primary and transformed data: the imported expression data is stored in an assay; additionally, the scaled data (arcsinh transformed) is also stored in an assay, meaning both can be easily accessed. Cell metadata: the first metadata added to this element of the structure will be a *sample_metadata* file imported by the user, containing any relevant metadata for the experiment, that is, days after treatment and graft versus host disease or none. Throughout the *ImmunoCluster* tool, more layers of metadata are added to cell metadata, that is, cell cluster identification (*FlowSOM* and *Phenograph*). Dimension reductions: dimensionality reduction coordinates, such as *UMAP* and *tSNE,* are stored and can be easily accessed throughout the *ImmunoCluster* tool for downstream analyses.

high-dimensional data are imported into the computational framework as FCS file formats or summarized feature expression values in CSV tabular format.

b. Associated metadata files need to be completed by the researcher in the experimental design stage (*Figure 7*). This includes a *sample_metadata* file, in which all metadata relating to samples will need to be input, such as timepoints, response to treatments, and patient status (*Figure 8—figure supplement 3A*). For simplicity in executing this step, the *panel_metadata* file allows users to rename parameters, such as the markers used in the analysis as well as select the markers that are to be used for different analyses stages (all markers may not need to be included in the following steps) (*Figure 8—figure supplement 3B*).

c. All of the above-mentioned data is stored within a SingleCellExperiment object (*Lun, 2017*). The SingleCellExperiment is an S4 class object that is implemented into the *ImmunoCluster* framework and is in essence a data container in which you can store and retrieve information such as metal-barcoding for sample multiplexing, metadata, dimensionality reduction coordinates, and more (outlined in *Figure 7*). FCS files can also be exported from the SingleCellExperiment using the *write_sce_to_fcs* function.

d. After the data and metadata files have been imported and stored in the SingleCellExperiment object, we provide several approaches for initial exploratory visualization of the data. One example of this is the MDS plots. The median marker expression data from each sample are used to create MDS plots, and therefore allow for an initial visualization of data points (labeled with metadata) and show the similarity and differences between samples in two-dimensions. Additionally, these plots can be used to detect batch effects and other technical artifacts such as antibody staining anomalies. Moreover, a heatmap of expression of markers measured for each patient can be created and metadata can be used to annotate the heatmap.

## Stage 2: Dimensionality reduction and unsupervised clustering

a. Dimensionality reduction is a key component of high-dimensional single-cell data analyses and enables researchers to view high-dimensional data. For example, the 33 protein markers analyzed by *Hartmann et al., 2019* can be reduced to lower dimensional embedded coordinates per cell. Nonlinear dimensionality reduction techniques can avoid overcrowding and represent data in distinct cell islands (*Figure 8*).

b. The framework offers users three nonlinear statistical methods for representing high-dimensional single-cell data in low-dimensional space (*Figure 8—figure supplement 4*). MDS (utilized in stage 1 for initial data exploration), uniform manifold approximation and projection (*UMAP*) (*Becht et al., 2019*), and t-Distributed Stochastic Neighbor Embedding (*tSNE*) (*van der Maaten and Hinton, 2008*) algorithms are available to generate a two-dimensional (2D) embedding of the data. However, the SingleCellExperiment object can store any other form of arbitrary dimensionality reduction, such as principal component analysis (*Blighe, 2020b*) or DiffusionMaps (*Angerer et al., 2016*), among others.

c. The density of cells within the distinct cell islands produced by the 2D embedding of multidimensional data can be viewed using the *density_plot* function (*Figure 2—figure supplement 1*).

d. After the data has been visualized in dimensionality reduced space, clustering algorithms can be used to identify cell communities within the data, which allows the identification of phenotype and abundance of cell clusters within populations/groups being analyzed. A number of clustering algorithms are available within the framework to define cell populations of interest in an unbiased manner. First, an ensemble (use of multiple algorithms) clustering method of *FlowSOM* (*Van Gassen, 2015b*) and Consensus clustering (*ConsensusClusterPlus* R package, *Wilkerson and Waltman, 2013*), which tests the stability of the clusters, leading to better results than applying a basic hierarchical clustering algorithm (*Weber and Robinson, 2016*). Additionally, the *PhenoGraph* clustering algorithm is available, which uses *K*-nearest neighbors

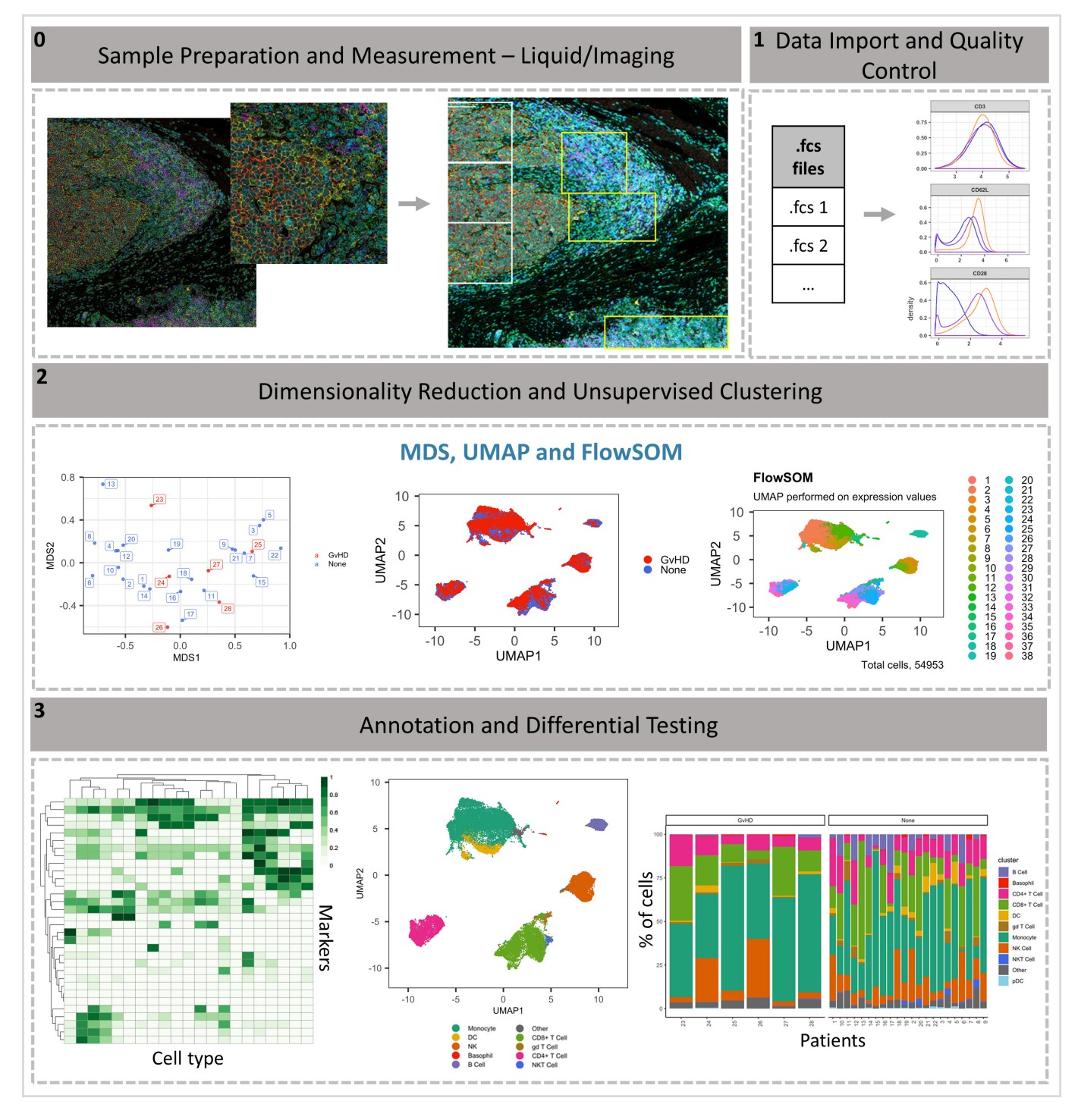

**Figure 8.** *ImmunoCluster* workflow overview. (**0**) Samples are stained/treated and measured, for imaging mass cytometry the tissue is segmented, and regions selected for further downstream analysis. After measurement, the raw data is normalized, concatenated, (combining flow cytometry standard files from the same samples, which may have been split due to large sample volume or technical issues) and gated, before importing into *ImmunoCluster*. (**1**) Quality control of data is carried out before analysis. (**2**) Data is reduced to two dimensions using either *UMAP* or *tSNE* algorithm, and data are clustered using the *FlowSOM* or *Phenograph* algorithms (these algorithms were selected as they are both high-performing unsupervised clustering algorithms; an in-depth comparison has previously been described by *Weber and Robinson, 2016*). (**3**) Data is visualized, and metadata, clusters, and cell-type labels are used to explore differences between samples/conditions. *A detailed step-by-step guide for using the *ImmunoCluster* tool is available: https://github.com/kordastilab/ImmunoCluster.

*Figure 8 continued on next page*

*Figure 8 continued*

The online version of this article includes the following figure supplement(s) for figure 8:

**Figure supplement 1.** Gating strategy for mass cytometry data in FlowJo.
**Figure supplement 2.** Imaging mass cytometry (IMC) data preprocessing workflow.
**Figure supplement 3.** Metadata files created in the experimental design stage.
**Figure supplement 4.** Dimensionality reduction.
**Figure supplement 5.** Selecting *K* clusters for visualization and downstream analysis.
**Figure supplement 6.** Elbow plot criterion to help determine optimal number of clusters for *K*-means clustering (*FlowSOM*).

and a Euclidean distance metric (*DiGiuseppe et al., 2018*). Lastly, *k-means clustering*, which classifies cells into clusters with high intra-class similarity and low inter-class similarity, where each cluster is represented by its center (centroid) that corresponds to the mean of points (e.g., markers) assigned to the cluster. Either or all clustering algorithms can be applied at the users' discretion.

e. The aim of the clustering algorithms is to assign all cells to *K* clusters ($K_1$, $K_2$, . . .), resulting in clusters corresponding to true cell types. Typically, a *K* larger than the number of expected cell types is chosen at this stage as the *ImmunoCluster* framework allows users to explore all *K* clusters (*Figure 8—figure supplement 5*), as well as collapsing clusters of the same cell type into one cluster if the user feels over-clustering has occurred. Over-clustering may enable the clustering identification of rare cell types, sometimes at the expense of creating several 'artificial' clusters of the more prevalent cell types. Additionally, the workflow allows the generation of an elbow plot that is created to help the user to select an appropriate number of clusters (*Figure 8—figure supplement 6*).

## Stage 3: Annotation and differential testing

a. After dimensionality reduction and clustering, the next step is to annotate each cluster based on marker expression. If over-clustering has occurred, multiple clusters of the same cell type can be present, and either a lower number of *K* clusters can be used; additionally, clusters of the same cell type can be collapsed into each other to create one cluster.

b. Clusters containing multiple cell types may also be identified; this could mean the user has under-clustered or occasionally a group of different cells are clustered together by the algorithms.

c. Expression of each marker analyzed can be projected onto the dimensionality reduced data by the researcher, aiding with the identification of cell types or phenotypically distinct clusters.

d. Outputs such as heatmaps showing the expression of all markers for each cluster can be used to identify the clusters of interest (*Figure 2—figure supplement 2*). These can be used to annotate figures produced in stage 3.

e. Biaxial dot plots with contouring of the SCE data can be produced using the *biaxial_plot* function; the expression of markers can be explored in the entire dataset or a subset of the data, that is, a particular cell cluster. A clustering option within *biaxial_plot* allows users to see the spread of data across two selected markers for different conditions (any metadata from the *sample_metadata* file), as well as different clusters (e.g., K1-60) (*Figure 3—figure supplement 1*).

f. Due to the type of data produced from IMC (lower resolution compared to LMC and FC), a novel function called *imc_rank* was specifically designed for the IMC data; this creates a heatmap in which the expression of each marker is ranked. This rank heatmap can be used to facilitate the identification of clusters that are 'high' or 'low' for particular markers to distinguish and logically assign cell populations manually.

g. These tools help define biological identity of cell clusters from stage 2 and construct a hierarchy of biologically meaningful cell populations based on marker expression and similarity.

h. The metadata added to the SingleCellExperiment can be used to annotate the dimensionality reduced (i.e., *UMAP*) and clustered (*FlowSOM*, *Phenograph, or K-means*) data, allowing for visualization of the distribution of cell islands between different condition/timepoints (*Figure 8*).

i. The above-mentioned metadata is also used to annotate all differential testing outputs, meaning the user can make many comparisons and explore multiple parameters all within the *ImmunoCluster* framework.

j.  A variety of differential testing outputs can be created, such as median marker expression for each sample (using clustered data) can be used to produce a hierarchical clustered heatmap, and box plots of cell cluster abundance between different conditions at different timepoints (any metadata can be used to annotate these figures).

k.  Statistically significant differences in the annotated populations between conditions are identified and displayed throughout the framework to the user for ongoing investigation and interpretation. Data from the *ImmunoCluster*-generated SingleCellExperiment object can be easily extracted for downstream analysis with differential analysis packages such as *diffcyt* (**Weber et al., 2019**) and *cydar* (**Lun et al., 2017**) for differential discovery, which is based on a combination of high-resolution clustering and empirical Bayes moderated tests. In addition, the *ImmunoCluster* framework offers the *stat_test_clust* and *stat_test_expression* functions that use the Wilcox rank-sum or Welch's T-test to identify which cell clusters are significantly different between conditions (any two-way comparison can be made using metadata input by user), and which markers are significantly different within cell clusters between different conditions, respectively.

## Acknowledgements

We would like to thank the following funding bodies for their support:

Guy's and St Thomas Hospital Charity: grant number: G170701, Jessica A Timms, Shahram Kordasti.

Cancer Research UK (CRUK), King's Health Partners Centre : Jessica A Timms, Shahram Kordasti C604/A25135.

Aplastic Anemia and MDS International Foundation (AAMDSIF): Shahram Kordasti.

Blood Cancer UK: Shahram Kordasti.

European Research Council (ERC): James N Arnold Start up grant 335326.

Medical Research Council (MRC): James W Opzoomer MR/N013700/1.

Medical Research Council (MRC): James W Opzoomer Doctoral Training Partnership in Biomedical Sciences.

Rosetrees Trust (Rosetrees): Sedigeh Kareemaghay, Mahvash Tavassoli M117-F2.

The funders had no role in study design, data collection and interpretation, or the decision to submit the work for publication. The research was supported by the Cancer Research UK King's Health Partners Centre and Experimental Cancer Medicine Centre at King's College London, and the National Institute for Health Research (NIHR) Biomedical Research Centre based at Guy's and St Thomas' NHS Foundation Trust and King's College London. The views expressed are those of the authors and not necessarily those of the NHS, the NIHR, or the Department of Health.

## Additional information

### Competing interests

Shahram Kordasti: Honoraria: Beckman Coulter, GWT-TUD, Alexion. Consulting or Advisory Role: Syneos Health. Research Funding: Celgene, Novartis pharmaceutical. The other authors declare that no competing interests exist.

### Funding

| Funder | Grant reference number | Author |
| --- | --- | --- |
| Cancer Research UK | C604/A25135 | Jessica A Timms<br>Shahram Kordasti |
| Aplastic Anemia and MDS International Foundation | | Shahram Kordasti |
| H2020 European Research Council | 335326 | James W Opzoomer<br>James N Arnold |
| Medical Research Council | MR/N013700/1 | James W Opzoomer |
| Medical Research Council | | James W Opzoomer |

| Rosetrees Trust | M117-F2 | Sedigeh Kareemaghay<br>Mahvash Tavassoli |
| Guy's and St Thomas' NHS<br>Foundation Trust | G170701 | Jessica Timms<br>Shahram Kordasti |
| LifeArc | 1118406 | Jessica Timms<br>Shahram Kordasti |

The funders had no role in study design, data collection and interpretation, or the decision to submit the work for publication.

### Author contributions

James W Opzoomer, Jessica A Timms, Conceptualization, Resources, Data curation, Software, Formal analysis, Validation, Investigation, Visualization, Methodology, Writing - original draft, Project administration, Writing - review and editing; Kevin Blighe, Conceptualization, Resources, Data curation, Software, Formal analysis, Validation, Investigation, Methodology, Writing - original draft, Writing - review and editing; Thanos P Mourikis, Software, Methodology, Writing - review and editing; Nicolas Chapuis, Data curation, Writing - review and editing; Richard Bekoe, Formal analysis; Sedigeh Kareemaghay, Paola Nocerino, Benedetta Apollonio, Mahvash Tavassoli, Investigation; Alan G Ramsay, Conceptualization, Resources, Supervision, Funding acquisition, Investigation, Methodology, Writing - original draft, Project administration, Writing - review and editing; Claire Harrison, Funding acquisition, Investigation, Writing - review and editing; Francesca Ciccarelli, Funding acquisition, Writing - review and editing; Peter Parker, Writing - review and editing; Michaela Fontenay, Investigation, Writing - review and editing; Paul R Barber, Resources, Software, Investigation, Writing - review and editing; James N Arnold, Conceptualization, Resources, Software, Supervision, Funding acquisition, Investigation, Methodology, Writing - original draft, Project administration, Writing - review and editing; Shahram Kordasti, Conceptualization, Resources, Supervision, Funding acquisition, Methodology, Writing - original draft, Project administration, Writing - review and editing

### Author ORCIDs

James W Opzoomer https://orcid.org/0000-0001-6842-756X
Jessica A Timms https://orcid.org/0000-0003-3687-9312
Alan G Ramsay http://orcid.org/0000-0002-0452-0420
Michaela Fontenay http://orcid.org/0000-0002-5492-6349
Paul R Barber http://orcid.org/0000-0002-8595-1141
Shahram Kordasti https://orcid.org/0000-0002-0347-4207

### Ethics

Human subjects: Formalin-fixed paraffin-embedded (FFPE) DLBCL tumor tissue was obtained from King's College Hospital, in accordance with the Declaration of Helsinki and approved by the UK National Research Ethics Committee (reference 13/NW/0040). Head and neck squamous cell carcinoma (HNSCC) tissue was obtained from King's College Hospital, consent was attained by the King Guy's & St Thomas' Research Biobank, within King's Health Partners Integrated Cancer Centre. The non-interventional study which collected bone marrow samples from elderly healthy donors was approved by the ethical committee of Cochin-Port Royal Hospital (Paris, France) (CLEP Decision N°: AAA-2020-08039).

### Decision letter and Author response

Decision letter https://doi.org/10.7554/eLife.62915.sa1
Author response https://doi.org/10.7554/eLife.62915.sa2

## Additional files

### Supplementary files

• Supplementary file 1. Reference panel of anti-human antibodies for mass cytometry used by *Hartmann et al., 2019*.

• Supplementary file 2. Reference panel of anti-human antibodies for the head and neck cancer (HNSCC) imaging mass cytometry experiment.

• Supplementary file 3. Reference panel of anti-human antibodies for the diffuse large B-cell lymphoma (DLBCL) imaging mass cytometry experiment.

• Supplementary file 4. Diffcyt computational framework output for differential discovery analysis for identified cell clusters in the *Hartmann et al., 2019* in LMC data.

• Transparent reporting form

## Data availability

The liquid mass cytometry dataset is available from FlowRepository (http://flowrepository.org/id/FR-FCM-Z244), the additional three datasets: 'Imaging mass cytometry data: Head and neck squamous cellcarcinoma tissue section', 'Flow cytometry data: healthy donor bone marrow taken during hip surgery', and 'Imaging mass cytometry data: Head and neck squamous cell carcinoma tissue section', are available from Dryad (https://datadryad.org/stash).

The following datasets were generated:

| Author(s) | Year | Dataset title | Dataset URL | Database and Identifier |
|---|---|---|---|---|
| Opzoomer JW, Timms J, Blighe K, Mourikis TP, Chapuis N, Bekoe R, Kareemaghay S, Nocerino P, Apollonio B, Ramsay AG, Tavassoli M, Harrison C, Ciccarelli F, Parker P, Fontenay M, Barber PR, Arnold JN, Kordasti S | 2021 | Imagingmass cytometry data: Head and neck squamous cell carcinoma tissuesection | https://doi.org/10.5061/dryad.gf1vhhmpr | Dryad, 10.5061/dryad.gf1vhhmpr |
| Opzoomer JW, Timms J, Blighe K, Mourikis TP, Chapuis N, Bekoe R, Kareemaghay S, Nocerino P, Apollonio B, Ramsay AG, Tavassoli M, Harrison C, Ciccarelli F, Parker P, Fontenay M, Barber PR, Arnold JN, Kordasti S | 2021 | Flow cytometry data: healthy donor bone marrow taken during hipsurgery | https://doi.org/10.5061/dryad.4b8gthtcf | Dryad, 10.5061/dryad.4b8gthtcf |
| Opzoomer JW, Timms J, Blighe K, Mourikis TP, Chapuis N, Bekoe R, Kareemaghay S, Nocerino P, Apollonio B, Ramsay AG, Tavassoli M, Harrison C, Ciccarelli F, Parker P, Fontenay M, Barber PR, Arnold JN, Kordasti S | 2021 | Imaging mass cytometry data: Diffuse large B-cell lymphoma lymph node section | https://doi.org/10.5061/dryad.3n5tb2rhd | Dryad, 10.5061/dryad.3n5tb2rhd |

The following previously published dataset was used:

| | | | | Database and |
|---|---|---|---|---|

| Author(s) | Year | Dataset title | Dataset URL | Identifier |
|---|---|---|---|---|
| Hartmann FJ, Babdor J, Gherardini PF, Amir ED, Jones K, Sahaf B, Marquez DM, Krutzik P, O'Donnell E, Sigal N, Maecker HT, Meyer E, Spitzer MH, Bendall SC | 2019 | Identification of disease-associated immune signatures following bone marrow transplantation | http://flowrepository.org/id/FR-FCM-Z244 | Flow Repository, FR-FCM-Z244 |

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

## Appendix 1

Staining protocol for a human tissue section taken from a patient with head and neck squamous cell carcinoma (HNSCC).

| Equipment | Consumables | Reagents | Reagent initial concentration | Reagent final Concentration |
|---|---|---|---|---|
| Humidified slide chamber (Sigma Aldrich, #H6644-1EA)* | Sterile 3 ml pastettes (Alpha Labs, #LW112) | DPBS, Ca, and Mg free (Gibco, #14190-094) | | |
| Dedicated glassware, coplin jars (Fisher, #10264732) ** | Superfrost Plus slides (Fisher, #ES-325) | Tween-20 (Sigma Aldrich, #P7949-100 ml)* | 100% | 0.1% |
| Orbital shaker | Kimberly-Clark Precision Wipers (Fisher, #10623111) * | Antigen retrieval solution pH 9.0 (R&D Systems, #CTS013) | | 1:10 in milli-Q water |
| Water bath | 0.2 syringe filter (TripleRed, #FPE204030) | Superblock (Thermo Fisher, #37515)** | | |
| Fridge | 50 ml syringes (Fisher, #10636531) | Iridium (Fluidigm, #201192B)*** | 500 µM | 0.5 µM in DPBS |
| Milli-Q water purification system (18.2 MΩ.cm) | Filter pipette tips | Metal tagged antibodies (Fluidigm) | | Require optimization for your tissue type |
| Hard slide box for storage in −80°C (VWR #631-1504) | PAP pen (Abcam, #ab2601) | FcR block (Trustain, Biolegend, # 422302) | 100% | 5% |
| Pipettes | | **Xylene** (Acros, #180862500) | | |
| Plastic tweezers, to handle slides (Farlamedical, #400200) | | **Ethanol** (Sigma Aldrich, #32221-2.5L) | | |
| Plastic coplin (Thermo Fisher, #E9611) **** | | BSA (Sigma Aldrich, #A9647) | 20% To prepare on the same day freshly | 10% |
| | | Kiovin Hu IgG Provide by Hospital's pharmacy | 10% 100 mg/ml | 5 mg/ml |

\* Alternatively, use a pipette tip box with moist towel at the bottom and piece of parafilm on the tray (this prevents evaporation and stops condensation gathering at the back of the slide, which can pool around edges and dilute antibodies).
\*\*Rinse in water, do not clean with detergent or in laboratory dishwasher (source of barium contamination).
\*\*\*\* Plasticware is required for 96°C water bath, used when dedicated histological autoclave or decloaking chamber is not available; you can also use a 50 ml centrifuge tube, loosely capped (Falcon or Corning).

\*Normal laboratory hand towels and blue roll are fine to tip excess buffer off slide surface or to place in the bottom of the humidified slide chamber. However, they should not be used to wipe the back of the slides or draw off excess buffer from front – as can transfer lint to the slides.

\*Permeabilization can also be conducted with Triton X100 or saponin
\*\* Alternative blocking solutions include BSA, human albumin, human immunoglobulin (KIOVIG from Pharmacy), serum same species as antibodies (mouse serum, Sigma Aldrich, # M5905-10ML).
\*\*\* DNA stain use alternative histone 3 antibody #3176023D.

## STAINING PROTOCOL DAY 1: SECTION CUTTING

Cut a 5 μm section from the block at and place in an incubation oven overnight (o/n) at 37˚C.
Cut a 3 μm section for hematoxylin and eosin (H&E) staining.

## STAINING PROTOCOL DAY 2:
## 1. PREPARATION

Prepare xylene and ethanol at 50:50 ratio ~ 50 ml of final volume.
Prepare the following ethanol dilutions, with milli-Q water: 96%, 90%, 80%, and 70% (~50 ml of final volume for each).
Dilute the 10X basic antigen retrieval buffer in milli-Q water in 50 ml falcon tube (leaving the cap loose) and heat to 96˚C in a water bath.
*BSA (Sigma Aldrich, #A9647): prepare a 20% solution fresh using, 0.4 g BSA in a plastic tube. Add 1 ml of superblock solution and incubate in the water bath for 1 hr. Once it is fully dissolved, top up to 2 ml (final volume).
Prepare the wash buffer using Dulbecco's phosphate buffered saline (DPBS) (calcium/magnesium free) and 0.1% Tween-20 (500 μl of Tween-20 [Sigma Aldrich, #P7949-100 ml] in the 500 ml DPBS bottle). Filter the wash buffer every time you use it with a 0.2 μm filter. Store at room temperature.

## 1. Notes for wash steps

Always conduct washes in ~50 ml of buffer in coplin jars on an orbital mixer.
Never wash in laboratory dishwasher or with detergent as this can represent a source of barium contamination. Also, do not use metal slide holders or forceps.

## 2. STAINING

Heat the 1X antigen retrieval solution (1:10 dilute in milli-Q water) to 96˚C in a water bath.
Heat a coplin jar and a bottle of wash buffer (0.1% Tween-20 in DPBS) to 37˚C in a water bath.
Prepare the humidified slide chamber (box with a damp tissue in the bottom and lid or piece of parafilm to cover).
Prepare blocking and antibody incubation buffer fresh: 10% BSA, 0.1% Tween-20, 1/20 dilution of FcR block (Trustain, Biolegend, #422302) and 1/20 dilution of KIOVIG (Kiovin Hu IgG) in Superblock (Thermo Fisher, #37515) solution. Use Superblock to top up to final volume. Avoid mixing, filtering, or centrifugation of the Superblock buffer prior to pipetting, let solution settle at 4˚C and pipette from top of the solution to avoid any aggregate material (that will have sunk to the bottom).

Conduct xylene washes in a safety cabinet.
Use separate coplin jars.
The xylene should be changed regularly, especially the first coplin jar in the sequence, which will have the most wax dissolved in it (readily become cloudy and less efficient).

## 1. CLEAR WAX FROM SLIDES (you sequentially remove the wax, with each wash in fresh xylene)

Incubate slide 1 hr at 60˚C.
Agitate slides (up and down motion) by hand, three washes at RT:
Xylene 100% for 5 min.

Xylene 100% for 5 min.
50:50 xylene:ethanol mix for 10 min.

## 1. REHYDRATE TISSUE (outside hood, use plastic tweezers)

Remove slides from xylene:ethanol mix in the hood, transfer to 100% ethanol for 5 min, and remove from cabinet.
Transfer slides to 90% ethanol – mix on orbital shaker for 5 min.
Transfer slides to 80% ethanol – mix on orbital shaker for 5 min.
Transfer slides to 70% ethanol – mix on orbital shaker for 5 min.
Remove slides and transfer to DPBS – mix orbital shaker for 10 min.
Slide can stay in DPBS until the antigen retrieval solution is ready.

## RETRIEVE ANTIGEN

Fluidigm have validated all their FFPE antibodies using 30 min antigen retrieval, with basic buffer, in a 96˚C water bath.

Heat 1X antigen retrieval solution to 96˚C in the water bath (in plastic coplin jar or closed falcon 50 ml centrifuge tube).
When the slides are added, the temperature may drop – do not start the timer until the probe indicates the temperature in the tube has returned to 96˚C.
Incubate slides in 1X antigen retrieval solution to 96˚C in the water bath (in closed 50 ml falcon centrifuge tube) and incubate for 30 min.
Cool down closed 50 ml falcon tube to around RT under cold running water.
Transfer slides to DPBS and wash on orbital shaker for 5 min.
Dry slides with tissue wipers (Thermo Fisher) and leave to dry for 10 s.
Draw a wax ring around the tissue.
Wash twice with filtered wash buffer (0.1% Tween-20) in DPBS on the orbital shaker for 8 min.
Filter it every time you use the wash buffer with 0.2 µm filter and store at RT.
Leave slides in the wash buffer until the block solution is ready.

## 1. BLOCK

While sections are in permeabilization buffer (0.1% Tween-20), prepare sufficient blocking buffer to cover the tissue, within the wax ring.
For a very small diameter wax ring, 50 µl will suffice; however, a large piece of tissue (>1 cm diameter) will require ~250 µl.

**Blocking buffer preparation**

| Solution | Dilute in | Initial concentration | Final concentration | Dilution factor | Per 100 µl | Per 300 µl |
|---|---|---|---|---|---|---|
| Tween-20 (Sigma Aldrich, #P7949-100 ml) | Superblock solution | Prepared first at 10% Then at 1% | 0.1% | 10 | 10 | 30 |
| Kiovin Hu IgG | Superblock solution | 10% 100 mg/ml | 5 mg/ml | 20 | 5 | 15 |
| FcR block (Trustain, Biolegend, # 422302) | Superblock solution | 100% | 5% (1/20) | 20 | 5 | 15 |
| BSA (Sigma Aldrich, #A9647) | Superblock solution | 20% (10 g in 100 ml = 10%) | 10% | 2 | 50 | 150 |
| Superblock (Thermo Fisher, #37515) | To top up Until your final volume | | | | 30 | 90 |

When you add the buffer to the inside of the wax circle, it should be contained and fully cover the tissue.

Remove slides from DPBS and remove excess buffer by tipping slide.
Use a lint-free tissue to dry the back of the slide thoroughly and to gently draw excess buffer away from the front, without touching the tissue or wax ring.
Transfer to humidified box, cover tissue with blocking buffer.
Incubate at RT for 90 min.

## 1. PREPARE ANTIBODY MIX AND INCUBATE

**Antibody mix**

| Cat# | Supplier | Tag | Target/ solution | Dil factor (1:X) | Initial concentration | Final concentration |
|------|----------|-----|------------------|-----------------|----------------------|---------------------|
| 3170019D | Fluidigm | 170 | CD3 (polyclonal, C-terminal) | 400 | | |
| 3156033D | Fluidigm | 156 | CD4 | 100 | | |
| 3152016D | Fluidigm | 152 | CD45 | 1000 | | |
| 3162035D | Fluidigm | 162 | CD8-alpha | 800 | | |
| 3161029D | Fluidigm | 161 | CD20 | 250 | | |
| MAB1561 | R&D Systems | 150 | PD-L1 | 50 | | |
| 3158029D | Fluidigm | 158 | E-cadherin | 1000 | | |
| 3159035D | Fluidigm | 159 | CD68 | 400 | | |
| 3165039D | Fluidigm | 165 | PD-1 | 50 | | |
| 3168022D | Fluidigm | 168 | Ki-67 | 400 | | |
| #A9647 | Sigma Aldrich | | BSA Prepare fresh | 2 | 20% | 10% |
| #422302 | Biolegend | | FCR | 20 | 100% | 5% |
| | Provided by hospital's pharmacy | | IgG Hu Kiovig | 20 | 100 mg/ml | 5 mg/ml |
| #P7949 | Sigma Aldrich | | Tween-20 | 10 | 1% | 0.1% |
| #37515 | Thermo Fisher | | Superblock | To top up to the required volume | | |

You will require the same volume of antibody mix used for blocking solution and tip off excess blocking buffer onto tissue paper.
Use a lint-free tissue to dry the back of the slide thoroughly and to gently draw excess buffer away from the front, without touching the tissue or wax ring.
Incubate slide o/n with antibody mix at 4°C in humidified chamber.

## STAINING PROTOCOL DAY 3
## 1. WASH IN PERMEABLIZATION BUFFER

Tip off the antibody mix onto tissue paper.
Flush the slide surface gently but thoroughly with a pastette and permeabilization buffer (DPBS with added 0.1% Tween-20) at RT.
Wash in the same RT buffer ~50 ml in coplin jar on the mixer for 8 min.
Repeat wash step for another 8 min.

## 1. WASH IN DPBS

In the same jar, decant 0.1% Tween-20 DPBS wash buffer and add further wash buffer at RT and then place on the mixer for 8 min.
Repeat wash step for another 8 min.

## 1. DNA STAIN

For initial testing, prepare iridium (Fluidigm) in DPBS at a final concentration of 0.5 µM (may require some optimization for your tissue type).

Dilute 125 µM iridium stock (Fluidigm 201192A) 1/250 with DPBS.

Tip off excess DPBS onto tissue paper.
Use a lint-free tissue to dry the back of the slide thoroughly and to gently draw excess buffer away from the front, without touching the tissue or wax ring.
Place slide in humidified chamber and load iridium solution into the wax ring so that it covers the tissue.
Incubate at RT for 30 min.
Flush the slide surface thoroughly with RT DPBS gently using a pastette.
Wash in RT DPBS on the mixer for 5 min.
Wash in milli-Q water ~50 ml in a clean coplin jar on the mixer for 5 min.
Air dry.

Slides can be stored for several months at RT in a cool, dry, dust-free storage container.

## Appendix 2

### Protocol for sectioning and staining of diffuse large B-cell lymphoma lymph node section

1. A 5 µm tissue section was deparaffinized with xylene and sequentially rehydrated in graded ethanol. Heat-induced antigen retrieval was performed in an automated pressure cooker (Menapath Antigen Access Unit, Menarini) at 125°C for 2 min in Antigen Retrieval Reagent-Basic (R&D Systems).
2. The tissue was then permeabilized with DPBS, 0.1% Tween-20 for 15 min, and blocked with Superblock (Thermo Fisher Scientific), 0.1% Tween-20, Fc Receptor Blocking solution (1:20; Biolegend; blocking buffer) for 2 hr at RT.
3. Antibodies were diluted together according to *Supplementary file 3* in blocking buffer and applied to the tissue overnight at 4°C.
4. After washing, the slide was incubated with the DNA intercalator (Cell-ID Intercalator-Ir, Fluidigm) for 30 min at RT.
5. The slide was then briefly washed with milli-Q water and air dried.
6. Acquisition of 1 mm$^2$ tissue region was carried out using the Hyperion imaging system (Fluidigm) at 200 Hz, with a resolution of 1 µm/pixel.

