## [Decision Letter]

**Acceptance summary:**

The revised manuscript now has a clearer analysis workflow. *ImmunoCluster* is a user-friendly pipeline R package for the analysis of flow and mass cytometry experiments and for imaging mass cytometry. We hope this provides non-computational immunologists with an opportunity to use dimensionality reduction, unsupervised clustering and differential expression/abundance analyses.

**Decision letter after peer review:**

Thank you for submitting your article "ImmunoCluster: A computational framework for the non-specialist to profile cellular heterogeneity in cytometry datasets" for consideration by *eLife*. Your article has been reviewed by 2 peer reviewers, and the evaluation has been overseen by a Reviewing Editor and Tadatsugu Taniguchi as the Senior Editor. The following individual involved in the review of your submission has agreed to reveal their identity: Evan W Newell (Reviewer #3).

The reviewers have discussed the reviews with one another and the Reviewing Editor has drafted this decision to help you prepare a revised submission.

The authors propose ImmunoCluster, a user-friendly pipeline R package for the analysis of flow and mass cytometry experiments and for imaging mass cytometry. The claim is that this approach and package is easy to use for non-computational immunologists and facilitates dimensionality reduction, unsupervised clustering, and differential expression/abundance analyses.

While this software is useful, the examples provided are good, there is very little/no novelty in the approach provided. The reviewers main concern is that the paper in its current form does not propose any new methodology with regards to clustering, visualization, or differential analysis. It was felt that the paper in its current form combines existing methodology in a user-friendly way that could indeed be important for biologists. In addition, the paper should acknowledge similar packages that already exist (eg CytofKit https://journals.plos.org/ploscompbiol/article?id=10.1371/journal.pcbi.1005112).

The combined reviewers' comments can be found below. For this manuscript to merit further consideration at *eLife*, clearer descriptions of ease of use of *Immunocluster* compared to other methods and other distinguishing features should be better highlighted and additional benchmarking comparisons.

Summary:

This tool enables advances but other tools already available that are sufficient and therefore the authors will need to convince a large portion of the researcher to use this tool, better comparisons should be provided to help in answering this question and then also better convince readers to use this software.

Essential revisions:

The authors note that several other similar packages influenced the development of *ImmunoCluster* (Refs 13-16) but it is unclear what makes this package novel. For instance, how does this workflow compare to other packages such as CATALYST (referenced by the authors) for quantifying cell cluster abundances and assessment of differential composition, which has been used in various ways in several publications (Crowell et al. F1000Res. 2020 [this one is new so not yet possible to have referenced], Weber et al., Communications Biology 2019, Nowicka et al., F1000Res. 2017, and related Fonseka et al. Sci. Trans. Med. 2018)? With regards to the differential abundance testing, there has been some excellent work in recent years, including DiffCyt (https://www.nature.com/articles/s42003-019-0415-5) and Cydar (https://pubmed.ncbi.nlm.nih.gov/28504682/). These methods among others seem to be the state-of-the-art in differential abundance analysis. How do your differential analysis results compare to these others, for example? Can you implement these methods in your pipeline?

The authors mention that the pipeline can accommodate millions of cells. Is this the case for all the clustering and visualization steps, or do you require downsampling? It would be helpful to see an analysis of run-time/details of the computer used to analyse the data. It appears the clustering methods implemented here (phenograph for example) might not work well if there are millions of cells across multiple samples.

In the MDS plots of samples (see Figure 3A), it is unclear what features are used to project the samples. Could you please elaborate on this point?

Does this current approach have any means for assessing or correcting for batch effects within the datasets? The authors only mention sample barcoding as an experimental plan to reduce batch effects, but fail to mention of any strategy to assess or correct for batch effects, which is a very important consideration that should be addressed.

Similarly, although channel spillover and differences in cell numbers were mentioned, no means were provided to assess or rectify these effects using the approach described.

---

## [Author Response]

Essential revisions:The authors note that several other similar packages influenced the development of ImmunoCluster (Refs 13-16) but it is unclear what makes this package novel. For instance, how does this workflow compare to other packages such as CATALYST (referenced by the authors) for quantifying cell cluster abundances and assessment of differential composition, which has been used in various ways in several publications (Crowell et al. F1000Res. 2020 [this one is new so not yet possible to have referenced], Weber et al., Communications Biology 2019, Nowicka et al., F1000Res. 2017, and related Fonseka et al. Sci. Trans. Med. 2018)? With regards to the differential abundance testing, there has been some excellent work in recent years, including DiffCyt (https://www.nature.com/articles/s42003-019-0415-5) and Cydar (https://pubmed.ncbi.nlm.nih.gov/28504682/). These methods among others seem to be the state-of-the-art in differential abundance analysis. How do your differential analysis results compare to these others, for example? Can you implement these methods in your pipeline?

We thank the reviewers for their comments. We have highlighted and discussed below where we have made revisions to address the comments raised. In response, we have added a new paragraph to the introduction that describes frameworks which influenced the development of *ImmunoCluster* and provide comparisons with other available frameworks, e.g. ***CATALYST***. We use this discussion to highlight and more clearly emphasize the unique aspects provided by the *ImmunoCluster* framework, such as the easily interpreted imaging mass cytometry data analysis, customizable figure outputs, and adaptive down-sampling. Each discussed in detail in the manuscript and below. We hope that these comparisons now more clearly define *ImmunoCluster*’s niche in this area.

**Influential packages and comparison with other available frameworks**

Lines 47-54: “**A number of previous computational cytometry workflows have been proposed, including *CyTOF workflow* [1], Diggins et al.[2] and *CapX*[3]. […] We have therefore developed *ImmunoCluster,* building on and extending the framework proposed in the *CyTOF workflow*[1] to create modular, flexible and easy-to-use implementations of cytometry analysis pipelines.”**

Lines 76-92: **“*ImmunoCluster* also contains functionality foradaptive down-sampling at the import and dimensionality reduction stages of the pipeline to overcome significant increases in runtime associated with large datasets in dimensionality reduction algorithms like UMAP and tSNE.[…] M**ethods for analyzing large cytometry datasets, in particular IMC datasets, in an open-source computational environment are currently limited.”

**Additional advantages**

We would also like to highlight below some of the key novel aspects of *ImmunoCluster* which are now emphasized within the manuscript.

Easily interpreted imaging mass cytometry data analysis: *ImmunoCluster* pipeline has the ability to carry out downstream imaging mass cytometry (IMC) data analysis. This allows users to run IMC data in line with flow cytometry (FC) and liquid mass cytometry (LMC) data analyses. Due to the lower resolution of IMC data (compared with FC and LMC) we specifically designed the ‘ranked expression heatmap’ function to work with IMC data, which allows users to rank markers measured to help identify cell populations. Presented in Figures 4 and 5.

lines 85-92: “**Importantly, and unique to *ImmunoCluster*, the scope of functionality in this package also enables the analysis of IMC data. […] M**ethods for analyzing large cytometry datasets, in particular IMC datasets, in an open-source computational environment are currently limited.”

Customizable figure outputs: In addition to the section mentioned above we have provided a new Supplementary Figure (Figure 1—figure supplement 1) to showcase our user-friendly ‘customizable figure outputs ’, detailing a variety of tailor-made figures produced by *ImmunoCluster*. We offer flexibility throughout the framework; researchers can easily adapt all figures and outputs to suit their specific needs. **The outputs from *ImmunoCluster* are created using the *ggplot* package which generates *ggplot* graphical objects that are flexible and easily modified**, resulting in an abundance of tailored outputs for the user to assess and use in reports, presentations, and publications. One example is the outputs a user can create from the dimensionality reduced data (e.g. UMAP algorithm). Users can project marker expression onto UMAP plots to help visually identify the cell types represented in the islands. The density of cells in each island can also be visualized using a density plot function. After clustering (e.g. FlowSOM), clusters can be projected onto the UMAP plot, the number of clusters can be easily changed so users can visually inspect different total number of clusters. Once users have assigned cell types, these can also be projected onto the UMAP plot and help users visually inspect their cluster assignments. Furthermore, researchers can use the metadata to colour and split the UMAP plots, allowing users to identify which cell islands are present in certain conditions or at particular time points (Figure 1—figure supplement 1).

**Lines 107-111: “*ImmunoCluster*** also offers user-friendly flexibility throughout the framework, researchers can easily adapt all figures and outputs to suit their specific needs resulting in an abundance of tailored outputs for the user to assess and use in publications, reports and presentations. An example detailing a variety of tailor-made figures produced by *ImmunoCluster* can be found in Figure 1—figure supplement 1.”

Adaptive down-sampling: *ImmunoCluster* also contains functionality for adaptive down-sampling at the import and dimensionality reduction stages of the pipeline to overcome significant increases in runtime associated with large datasets in dimensionality reduction algorithms like UMAP and tSNE.

Lines 76-78: “***ImmunoCluster* also contains functionality for adaptive down-sampling at the import and dimensionality reduction stages of the pipeline to overcome significant increases in runtime associated with large datasets in dimensionality reduction algorithms like UMAP and tSNE.”**

**Differential abundance testing**

The *ImmunoClusterstat_test_clust* function allows users to carry out t-test and Wilcoxon rank sum test and export results tables which we feel enrich the utility of the pipeline for the basic research scientist, who will now be able to more easily analyze and interrogate statistically significant biological changes within the data.

In response to the reviewers’ comment regarding differential abundance testing, we carried out a concurrent test using the *diffCyt* framework with the Hartmann et al.[9] data to benchmark the output of the pipeline with currently available ‘gold standard’ tools. Additionally, we highlight that users can easily extract the data from the*ImmunoCluster* generated SingleCellExperiment object and carry out downstream analyses with other packages such as *diffcyt.* Text now reads:

Lines 172-179: “We compared *ImmunoCluster’s* differential abundance testing output (*stat_test_clust*), run in the t-test mode, with the Hartmann et al. data[9] into the*diffcyt* computational framework [6], which is a state-of-the-art tool fordifferential discovery analyses. *Diffcyt* identified the same three cell clusters (memory B-cells, naïve B-cells and naïve CD4^+^ T-cells) as differentially abundant (FDR p<0.05) between the GvHD and none conditions. This demonstrates concordance with *ImmunoCluster’s* statistical output that identified naïve B-cells and naïve CD4^+^ T-cells as one of the principle differentially abundant populations, which was also identified in the original analysis of the data (Supplementary file 4).”

Lines 488-495: “Data from the *ImmunoCluster* generated SingleCellExperiment object can be easily extracted for downstream analysis with differential analysis packages such as *diffcyt* [6] and *cydar* [10] for differential discovery, which is based on a combination of high-resolution clustering and empirical Bayes moderated tests. In addition, the *ImmunoCluster* framework offers the *stat_test_clust* and *stat_test_expression* functions which use the Wilcox rank sum or Welch’s T-test to identify which cell clusters are significantly different between conditions (any two-way comparison can be made using metadata input by user), and which markers are significantly different within cell clusters between different conditions, respectively.”

The authors mention that the pipeline can accommodate millions of cells. Is this the case for all the clustering and visualization steps, or do you require downsampling? It would be helpful to see an analysis of run-time/details of the computer used to analyse the data. It appears the clustering methods implemented here (phenograph for example) might not work well if there are millions of cells across multiple samples.

The *ImmunoCluster* package contains functionality to adaptively down-sample cell number at both the data import and dimensionality reduction stage. For computationally intensive dimensionality steps such as *UMAP* and *tSNE* down-sampling is likely required for large datasets with >1Million cells. Down-sampling is not required for clustering depending on the method selection, as suggested *Phenograph* will not scale effectively, but the *FlowSOM* method implemented will scale to large datasets. We do not require down-sampling for any of the additional visualization steps for the large 2.3 Million cell published LMC GvHD dataset analyzed. In response to the reviewer’s comment we have now provided a new Supplementary Figure with runtimes for a typical pipeline generated using *ImmunoCluster* and at different levels of down-sampling (Figure 1—figure supplement 2). We have also modified the text to read:

Lines 124-126: “We observed that a typical pipeline run on the full 2.3 million cell LMC GvHD dataset (with UMAP down-sampling to 500k cells) would take approximately 110 minutes on a 2.9 GHz Intel Core i7 Macbook pro with 16 GB RAM (Figure 1—figure supplement 2).”

Lines 504-514: “The inclusion of adaptive down-sampling of cells for the running of computationally intensive dimensionality reduction steps, using tools such as *UMAP* and *tSNE*, allows *ImmunoCluster* to generate cytometry analysis pipelines that readily scalable to a dataset of millions of cells across several samples and a variety of experimental or phenotypic conditions. ***[…]*** Whilst the *PhenoGraph* clustering method does not readily scale, the *FlowSOM*-based clustering method can be implemented on much larger datasets.”

In the MDS plots of samples (see Figure 3A), it is unclear what features are used to project the samples. Could you please elaborate on this point?

Many thanks for spotting this important point, our apologies this wasn’t clear. In response, the median marker expression data in each sample were used to create the MDS plot, we have added this clarification in both the text and figure legend.

Lines 408-412: “One example of this is the **Multidimensional scaling (MDS) plots.The median marker expression data from each sample are used to create MDS plots, and therefore allow for an initial visualization of data points (labelled with metadata) and show the similarity and differences between samples in two-dimensions.”**

Lines 617-621: “Figure 1. Initial exploration of LMC data from patients with leukemia who received bone marrow transplants. (A-B) Multidimensional scaling (MDS) of data, median marker expression data from each sample were used to create plots, annotated with condition (GvHD or none) (A), and time after bone marrow transplant (BMT) treatment (B). C Heatmap showing **the median marker expression for each patient.”**

Does this current approach have any means for assessing or correcting for batch effects within the datasets? The authors only mention sample barcoding as an experimental plan to reduce batch effects, but fail to mention of any strategy to assess or correct for batch effects, which is a very important consideration that should be addressed.

We agree with the reviewer’s comment that assessing and correcting for batch effects is very important, therefore in response we have now added a specific paragraph discussing strategies used for batch correction to the manuscript which will further guide and support the utilization of *ImmunoCluster*. Text now reads:

Lines 282-292: “Technical inter-sample marker signal variability due to batch effects may impact the ability of unsupervised analysis to reliably detect certain populations, if significant batch effects are present. **[…]** Statistical methods for batch correction may also be applied, in recent years a class of methods called Remove Unwanted Variation (RUV) have been developed, and *CytofRUV* [12] is a recently developed package specifically designed for CyTOF dataset batch correction.”

Similarly, although channel spillover and differences in cell numbers were mentioned, no means were provided to assess or rectify these effects using the approach described.

In response, we have also now expanded our description on how to rectify channel spillover within the text.

Lines 298-304: “Although channel spillover is diminished in mass cytometry, it still exists in fluorescence-based FC, and should be considered when designing antibody panels to reduce the effects on introducing cell phenotype artifacts in downstream unsupervised analysis. […] where cells are stained with all fluorescently-tagged antibodies except for the one of interest [13].”

Furthermore, we have also expanded our descriptions around the potential issues associated with cell number disparity between samples.

Lines 292-298: “If there are significant differences in the total number of cells recovered between samples, samples with many more cells may bias the clustering. […] If a particular sample has very few cells they may need to be excluded from the analysis as they may not be a representative sample.”

**References:**

https://github.com/HelenaLC/CATALYST

https://github.com/kevinblighe/scDataviz

1. Nowicka, M., et al., CyTOF workflow: differential discovery in high-throughput high-dimensional cytometry datasets [version 4; peer review: 2 approved]. 2019. 6(748).2. Diggins, K.E., P.B. Ferrell, Jr., and J.M. Irish, Methods for discovery and characterization of cell subsets in high dimensional mass cytometry data. Methods (San Diego, Calif.), 2015. 82: p. 55-63.3. Ashhurst, T.M., et al., Analysis of the Murine Bone Marrow Hematopoietic System Using Mass and Flow Cytometry, in Mass Cytometry: Methods and Protocols, H.M. McGuire and T.M. Ashhurst, Editors. 2019, Springer New York: New York, NY. p. 159-192.4. Crowell H, Z.V., Chevrier S, Robinson M, CATALYST: Cytometry dATa anALYSis Tools. R package version 1.12.2, . 2020.5. Chen, H., et al., Cytofkit: A Bioconductor Package for an Integrated Mass Cytometry Data Analysis Pipeline. PLOS Computational Biology, 2016. 12(9): p. e1005112.6. Weber, L.M., et al., diffcyt: Differential discovery in high-dimensional cytometry via high-resolution clustering. Communications Biology, 2019. 2(1): p. 183.7. Ashhurst, T.M., et al., Integration, exploration, and analysis of high-dimensional single-cell cytometry data using Spectre. bioRxiv, 2020: p. 2020.10.22.349563.8. Blighe, K., “scDataviz: single cell dataviz and downstream analyses.” . 2020.9. Hartmann, F.J., et al., Comprehensive Immune Monitoring of Clinical Trials to Advance Human Immunotherapy. Cell Rep, 2019. 28(3): p. 819-831.e4.10. Lun, A.T.L., A.C. Richard, and J.C. Marioni, Testing for differential abundance in mass cytometry data. Nature Methods, 2017. 14(7): p. 707-709.11. Schulz, A.R. and H.E. Mei, Surface Barcoding of Live PBMC for Multiplexed Mass Cytometry. Methods Mol Biol, 2019. 1989: p. 93-108.12. Trussart, M., et al., Removing unwanted variation with CytofRUV to integrate multiple CyTOF datasets. *eLife*, 2020. 9: p. e59630.13. Roederer, M., Spectral compensation for flow cytometry: Visualization artifacts, limitations, and caveats. 2001. 45(3): p. 194-205.